# A positive feedback loop bi-stably activates fibroblasts

So-Young Yeo[1], Keun-Woo Lee[1], Dongkwan Shin[2], Sugyun An[2], Kwang-Hyun Cho[2] & Seok-Hyung Kim[1,3,4]

Although fibroblasts are dormant in normal tissue, they exhibit explosive activation during wound healing and perpetual activation in pathologic fibrosis and cancer stroma. The key regulatory network controlling these fibroblast dynamics is still unknown. Here, we report that Twist1, a key regulator of cancer-associated fibroblasts, directly upregulates Prrx1, which, in turn, increases the expression of Tenascin-C (TNC). TNC also increases Twist1 expression, consequently forming a Twist1-Prrx1-TNC positive feedback loop (PFL). Systems biology studies reveal that the Twist1-Prrx1-TNC PFL can function as a bistable ON/OFF switch and regulates fibroblast activation. This PFL can be irreversibly activated under pathologic conditions, leading to perpetual fibroblast activation. Sustained activation of the Twist1-Prrx1-TNC PFL reproduces fibrotic nodules similar to idiopathic pulmonary fibrosis in vivo and is implicated in fibrotic disease and cancer stroma. Considering that this PFL is specific to activated fibroblasts, Twist1-Prrx1-TNC PFL may be a fibroblast-specific therapeutic target to deprogram perpetually activated fibroblasts.

[1] Department of Health Science and Technology, Samsung Advanced Institute for Health Science and Technology, Sungkyunkwan University, Seoul 06351, Republic of Korea. [2] Department of Bio and Brain Engineering, Korea Advanced Institute of Science and Technology (KAIST), Daejeon 34141, Republic of Korea. [3] Department of Pathology, Samsung Medical Center, Sungkyunkwan University School of Medicine, Seoul 06351, Republic of Korea. [4] Single Cell Network Research Center, Sungkyunkwan University School of Medicine, Suwona, Gyeonggi-do 16419, Republic of Korea. These authors contributed equally: So-Young Yeo, Keun-Woo Lee, Dongkwan Shin. Correspondence and requests for materials should be addressed to K.-H.C. (email: ckh@kaist.ac.kr) or to S.-H.K. (email: platoshkim@daum.net)

Fibroblasts play a pivotal role in organ development and tissue repair through secreting and remodeling the extracellular matrix (ECM), and also in regulating epithelial differentiation through mesenchymal−epithelial crosstalk[1,2]. In healthy and intact tissue, fibroblasts remain quiescent and nonproliferative; when tissue is injured, the explosive expansion in the fibroblast population is observed in the early proliferative phases of wound healing[3]. Once the wound is repaired, the number of fibroblasts markedly decreases owing to massive apoptosis[4,5]. A question then arises regarding what process is responsible for the fibroblasts' proliferative burst and subsequent tapering off. Under pathologic conditions due to unknown mechanisms, fibroblast regression does not occur, which can lead to devastating fibrotic diseases including IPF (idiopathic pulmonary fibrosis), keloid disease, systemic sclerosis, desmoid tumor, and chronic obstructive pulmonary disease[2,6]. Fibroblast-related disorders driven by perpetual fibroblast activation also include cancer because fibroblasts in cancer stroma strongly promote cancer progression by orchestrating cancer-promoting inflammation[5,7].

An insight into the mechanism of fibroblast activation can be obtained by considering the basic features of fibroblast activation. Unlike constantly dividing cells such as epithelial cells, fibroblasts are normally quiescent and activated only when needed[5]. This strongly suggests that there are two distinct phenotypes in fibroblast: quiescent and activated. This indicates that fibroblasts possess a bistable switch circuit that can shift from one stable state to another without resting in an intermediate state. In fact, all-or-none bistability is a key for understanding the decision-making processes in a number of cellular functions including cell cycle progression, cellular differentiation, and apoptosis[8,9]. Positive feedback loops (PFLs) are known to generate such all-or-none bistable responses by converting continuously graded inputs into discrete outputs[9,10]. This led us to hypothesize that some sort of PFL establishes the bistable nature of fibroblast activation and identifying the motifs constituting such PFLs should be a key undertaking of fibroblast biology.

To understand in detail which molecules form such PFLs in fibroblasts, we first explored a key regulator of fibroblast activation. Previously, we discovered that Twist1 is highly expressed in cancer-associated fibroblasts (CAFs) but not in normal quiescent fibroblasts[11–13]. Twist1 is a basic helix-loop-helix transcription factor that is essential for the development of mesodermally derived tissues[14,15]. Through further study, we identified that Twist1 is able to trans-differentiate normal quiescent fibroblasts to CAFs[13]. Additionally, Twist1 was reported to inhibit the senescence and apoptosis of fibroblasts[13,16]. Furthermore, Twist1 was reported to be highly expressed in perpetually activated, abnormal fibroblasts of IPF[16]. All of these findings strongly suggest that Twist1 can be considered a clue to unveiling such PFL that is responsible for the bistable characteristic of fibroblast activation.

Therefore, in this study, we started our quest for such PFL by investigating downstream targets of Twist1 in fibroblasts, and we discovered a Twist1-centered PFL that regulates fibroblasts activation. This PFL, comprising Twist1, Prrx1 (paired-related homeobox 1), and tenascin-C (TNC), can switch nonproliferative phenotypes to proliferative ones. Prrx1 functions as a transcription factor and regulates mesenchymal differentiation during embryonic development, much like Twist1[17]. Interestingly, a recent study revealed that both Twist1 and Prrx1 are major members of the fibroblast-specific key transcriptional network[18]. TNC is a glycoprotein that forms an ECM during development and tissue injury. TNC is highly expressed during embryogenesis, whereas its expression is absent or found only in trace amounts in developed organs[19]. TNC has been shown to be upregulated under pathological conditions caused by inflammation, infection, and tumorigenesis[19,20].

Through systems biology analysis combining mathematical simulation with biochemical experiments, we found that this PFL is implicated not only in physiological activation of fibroblast but also in the irreversible constitutive activation observed in pathologic fibroblasts. In particular, we found that the injection of exogenous TNC into the wound sites of wild-type mice induced hyperactivation of this PFL, then generated multifocal fibroblastic nodules histologically similar to IPF. This fibroblastic nodule was not observed in fibroblast-specific Twist1 knockout mice despite exogenous TNC injection. In addition, in a clinical study, we observed that these three genes were frequently coexpressed in "fibroblast hot spots" of pathologic fibrotic diseases and cancer stroma in various carcinomas (epithelial cancers). These results suggest that Twist1-Prrx1-TNC PFL may be a core component of bistable (ON/OFF) switch determining fibroblast activation in wound healing and various fibrotic diseases.

## Results

**Twist1 is a potent but indirect inducer of TNC in CAFs**. In our previous study, we found that TNC is a strong candidate for a downstream target of Twist1 in CAFs through mRNA microarray-based analysis[13]. Because TNC plays a crucial role in embryonic mesenchyme, wound healing, and cancer stroma[21], we suspected that the Twist1-TNC axis may confer normal fibroblasts with CAF-like properties. First, we found that upregulation of Twist1 expression in normal fibroblasts strongly induced TNC expression in both protein and mRNA levels (Fig. 1a). In addition, silencing of Twist1 in CAFs, meanwhile, significantly downregulated TNC expression (Fig. 1a). Next, we examined whether Twist1 directly regulates the transcription of TNC, as a transcription factor. Although Twist1 was added, the TNC promoter activity was not increased (Fig. 1b). Furthermore, chromatin immunoprecipitation (ChIP) assay revealed that Twist1 did not bind to any of the E-box sequences within the TNC promoter in two CAFs (Fig. 1c). These results strongly suggest that Twist1 induces TNC expression indirectly via a third factor.

Although TNC has been known to be produced by fibroblasts[21], its effect on the fibroblasts themselves has rarely been studied. Indeed, we found that downregulation of TNC led to decreased fibroblast proliferation, denoted by reduced colony formation (Fig. 1d and Supplementary Fig. 1). The ECM remodeling capacity was also decreased by the silencing of TNC (Fig. 1d). Additionally, the apoptosis of fibroblasts was enhanced by the loss of TNC (Fig. 1d). These findings indicate that TNC is not only produced from fibroblasts but also is essential for their activation; therefore, we suspected that Twist1 expression was also influenced by TNC in a feedback pattern. Indeed, exogenous TNC alone increased Twist1 expression in normal fibroblasts, whereas downregulation of TNC decreased Twist1 expression in CAFs (Fig. 1e). TNC-induced Twist1 overexpression was attenuated by LM609 (Integrin β3 blocking Ab) treatment, indicating that upregulation of Twist1 by TNC was mediated by Integrin β3 (Fig. 1f). Therefore, TNC is a necessary and sufficient condition for Twist1 expression in fibroblasts (Fig. 1g).

**Twist1 is a positive regulator of Prrx1 in fibroblasts**. To search for a candidate gene that mediates the Twist1-induced TNC expression, we analyzed the external cell type-specific gene expressional database FANTOM5, which contains data from 988 human primary cells and tissues; we found that 11 genes were coexpressed with Twist1 and TNC simultaneously in both fibroblasts and stem cells (Fig. 2a). Of these 11 genes, Prrx1 has been previously reported to increase TNC expression during

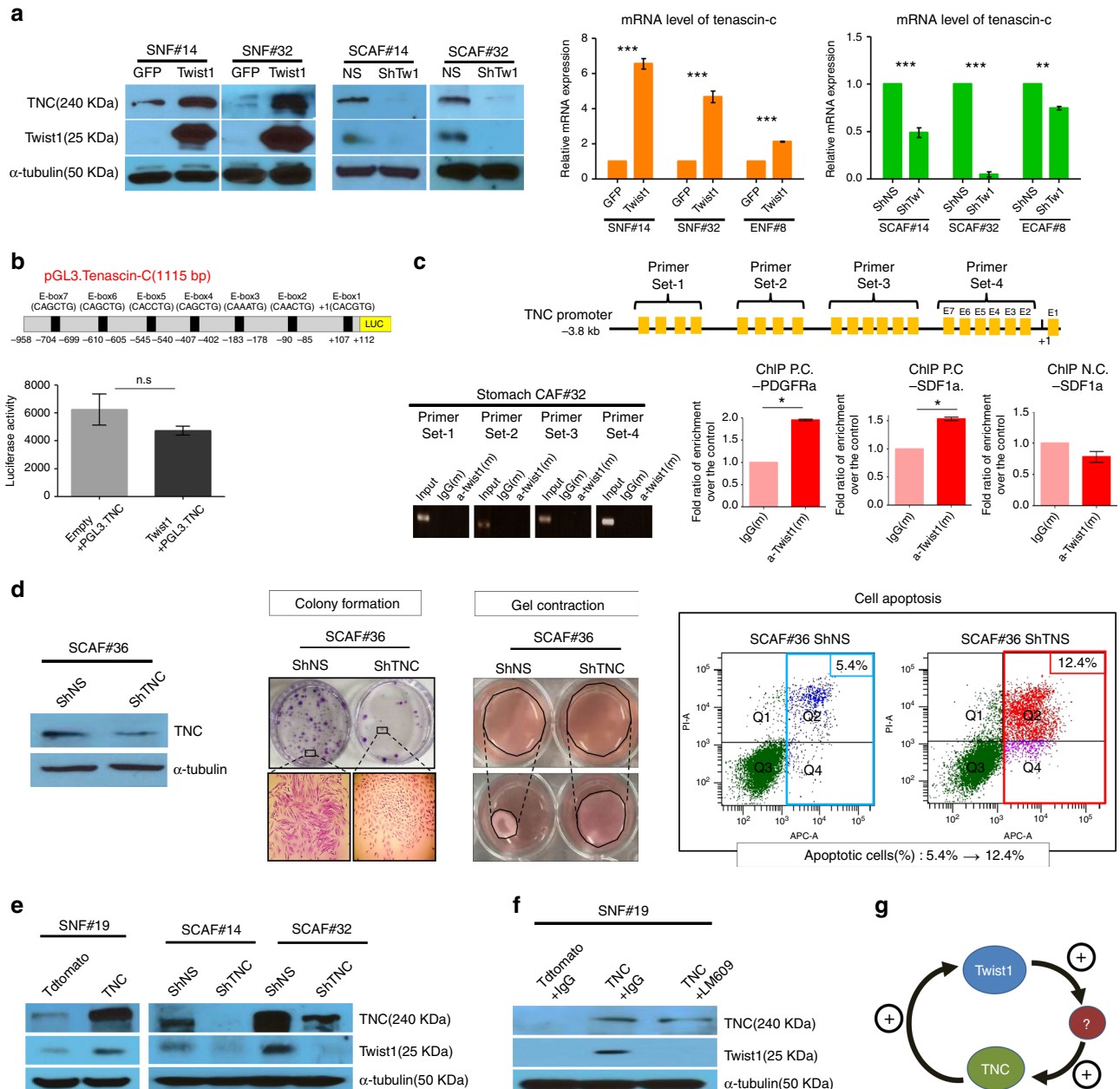

**Fig. 1** Twist1 increases tenascin-c expression in cancer-associated fibroblasts. Twist1 is a potent but indirect inducer of tenascin-c (TNC), which is essential for maintaining Twist1 expression in cancer-associated fibroblasts (CAFs). **a** Normal fibroblasts were transduced with Twist1 encoding lentivirus or GFP control. Induced Twist1 expression increased both the protein and mRNA of TNC in stomach normal fibroblasts (SNF#14 and SNF#32) and esophageal normal fibroblasts (ENF#8). CAFs were transduced with lentivirus-expressing shRNA specific for Twist1 (shTw1) or nonspecific shRNA (shNS). Knockdown of Twist1 also decreased the TNC expression in the esophageal and stomach cancer-derived CAFs (ECAF#8, SCAF#14, and SCAF#32). Real-time RT-PCR data are presented as the mean ± SEM; $N = 3$ independent experiments (two-tailed $t$ test: **$p<0.001$, ***$p<0.0001$). **b** We transfected HT1080 cells with TNC promoter-driven ($-938 \sim +112$bp) luciferase reporter genes simultaneously with a Twist1 expression vector. Although we added Twist1, the luciferase reporter assay revealed that Twist1 did not enhance the TNC promoter activity. Data are presented as the mean ± SEM; $N = 4$ independent experiments (two-tailed $t$ test: ns not significant). **c** The chromatin immunoprecipitation assay indicated that Twist1 did not bind to any E-box sequences between $-3.8$kb and $+25$bp of the TNC transcription start sites. In contrast, Twist1 bound to independent positive controls such as E-box sequences in the PDGFRa and SDF1a promoter (ChIP P.C-PDGFRa / SDF1a) but not to negative control, non E-box sequence in SDF1a promoter (ChIP N.C.-SDF1a). Data are presented as the mean ± SEM; $N = 3$ independent experiments (two-tailed $t$ test: *$p<0.05$). **d** The shRNA-mediated depletion of TNC led to decreased colony formation and gel contraction abilities in stomach cancer-derived CAFs (SCAF#36). The TNC silencing also increased the proportion of apoptotic cells. **e** Exogenous TNC expression in normal fibroblasts (SNF#19) enhanced the Twist1 expression, whereas the TNC depletion in stomach cancer-derived CAFs (SCAF#14 and #32) led to Twist1 downregulation. **f** TNC-induced Twist1 expression was mediated by Integrin β3. LM609 (Integrin β3 blocking Ab) abrogated TNC-mediated Twist1 expression in normal fibroblasts (SNF#19). **g** Schematic diagram of the feedback regulation between Twist1 and TNC

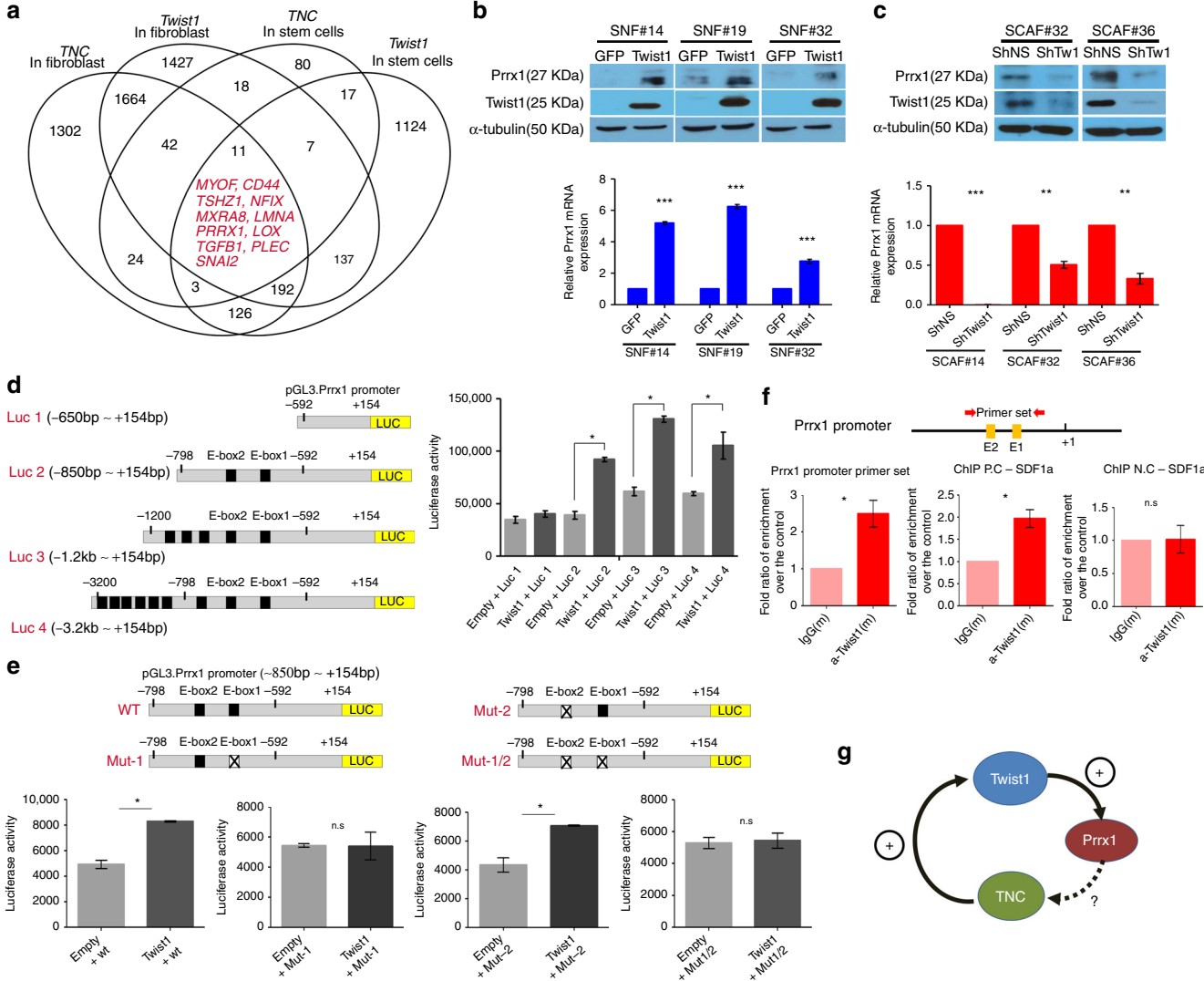

**Fig. 2** Twist1 was an essential, positive transcriptional regulator of Prrx1 expression in both normal fibroblasts and CAFs. Twist1 enhanced the expression of Prrx1, a potential TNC inducer, by direct transcriptional activation. **a** Analysis of the FANTOM database revealed 11 candidate genes whose expressions were significantly correlated with both Twist1 and TNC expression in fibroblasts and stem cells. **b** Induction of Twist1 expression increased Prrx1 expression in normal fibroblasts. **c** The knockdown of Twist1 expression in CAFs led to decreased Prrx1 expression. Data are presented as the mean ± SEM; $N = 3$ independent experiments (two-tailed $t$ test: $**p<0.001$, $***p<0.0001$). **d** To determine whether Twist1 regulates Prrx1 promoter activity, Prrx1 promoter-driven luciferase gene and Twist1 expression vector were cotransfected into HT1080 fibrosarcoma cells. The luciferase reporter assay revealed that the ectopic expression of Twist1 increased the Prrx1 promoter activity. Of the four different Prrx1 promoter constructs, construct #2 containing two E-boxes between −798 and −592 bp from the TSS of Prrx1 showed the most obvious difference in response to transient cotransfection with a Twist1 expression vector. Data are presented as the mean ± SEM; $N = 3$ independent experiments (two-tailed $t$ test: $*p<0.001$). **e** Two E-box mutations in E-box 1 (CATCTG (−619) to GGTCTG) and E-box 2 (CAACTG (−787) to TCACTG) were introduced in HT1080 cells. The Twist1-induced Prrx1 promoter activity was negated by the E-box mutation that was located at 619 bp of the TSS of Prrx1. Data are presented as the mean ± SEM; $N = 3$ independent experiments (two-tailed $t$ test: $*p<0.05$). **f** The schematic illustration of the ChIP assay PCR primer binding sites in the Prrx1 promoter region. We used the E-box sequence in the SDF1a promoter as a positive control. Compared with the IgG and positive controls, the Prrx1 promoter region that contained E-box 1 and E-box 2 was specifically enriched by each primer set after chromatin immunoprecipitation. That is, the ChIP assay revealed that Twist1 binds to the Prrx1 promoter sites (−819 to −597 bp) that contain two E-boxes. Data are presented as the mean ± SEM; $N = 3$ independent experiments (two-tailed $t$ test: $*p<0.05$). **g** Schematic diagram of the regulatory relationship among Twist1, Prrx1, and TNC

embryonic development[22]. Therefore, we suspected that Twist1 induces the upregulation of TNC by Prrx1, and indeed, Twist1 upregulation in normal fibroblasts induced Prrx1 expression at both the protein and mRNA levels (Fig. 2b). shRNA-mediated Twist1 repression in CAFs led to the downregulation of both the Prrx1 protein and mRNA (Fig. 2c). In addition, Twist1 increased the Prrx1 promoter activity, and the sequences between −850 and −650 bp upstream of the Prrx1 promoter transcription start site were the most critical for this enhanced promoter activity by

Twist1 (Fig. 2d). This region contains two candidate E-box sequences for Twist1 binding. To determine which of these two was a Twist1 binding site, we performed a combined mutagenesis and promoter-reporter assay. Mutation of the proximal E-box (−619) abrogated the Twist1-induced Prrx1-promoter activity, whereas mutation of the distal E-box (−787) made no difference; this suggested that the proximal E-box is essential for the Twist1-induced Prrx1 promoter activity (Fig. 2e). Furthermore, a ChIP assay confirmed that Twist1 physically binds to this region

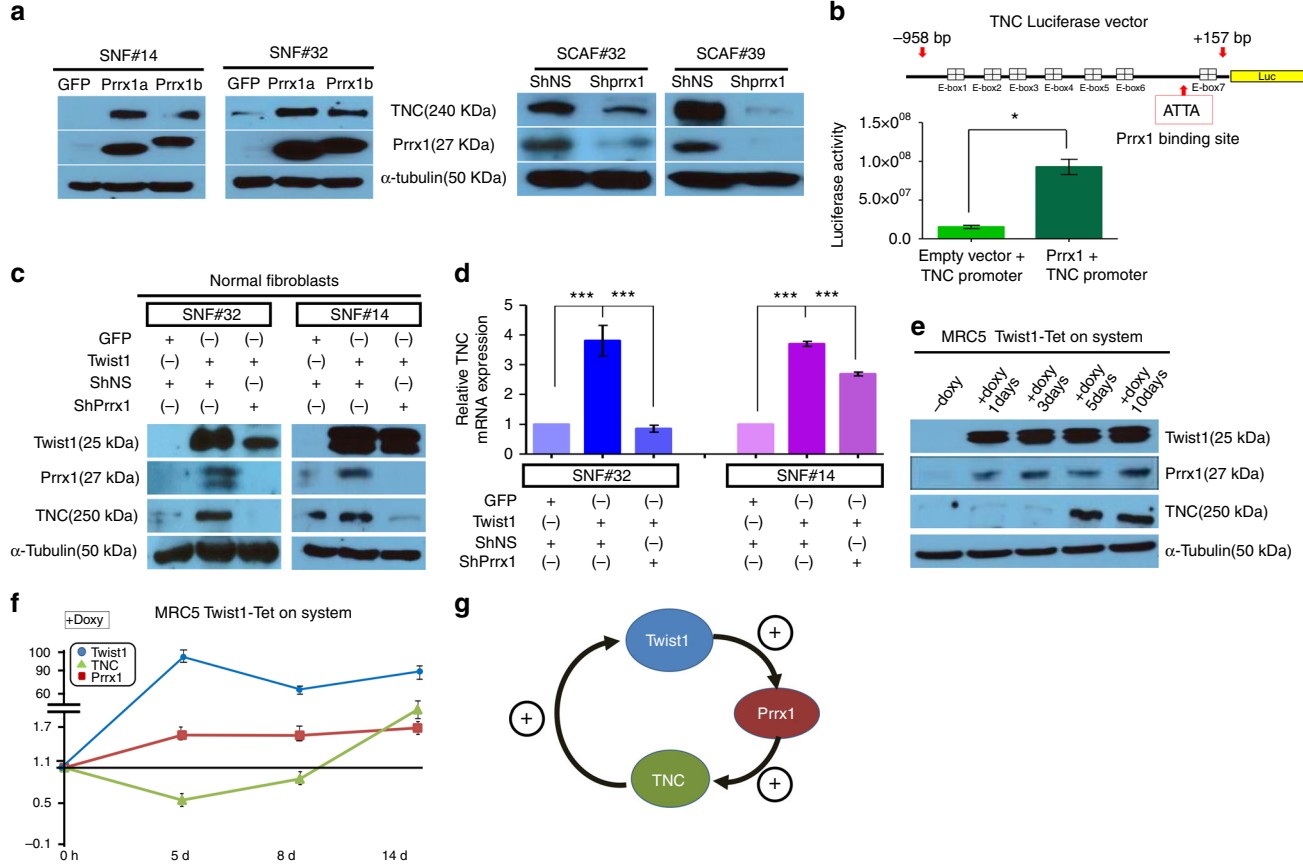

**Fig. 3** Twist1, Prrx1, and TNC formed a positive feedback loop in fibroblasts. **a**, **b** Prrx1 directly upregulates TNC expression in fibroblasts. **a** Both Prrx1a and Prrx1b increased the TNC protein level in normal fibroblasts (SNF#14 and #32). Prrx1 depletion led to the downregulation of TNC expression in CAFs (SCAF#32 and #39). **b** Prrx1 enhanced the TNC promoter activity. Data are presented as the mean ± SEM; $N = 3$ independent experiments (two-tailed $t$ test: *$p<0.01$). **c**, **d** Prrx1 is indispensable for Twist1-induced TNC expression at the protein (**c**) and mRNA (**d**) level. Data are presented as the mean ± SEM; $N = 4$ independent experiments (two-tailed $t$ test: ***$p<0.0001$). **e**, **f** To confirm the Twist1-Prrx1-TNC PFL, we introduced a doxycycline-inducible Twist1 expression vector into two fibroblast cell lines (MRC5 and IMR90) as well as a normal fibroblast line (SNF#19). Induced Twist1 expression using this Tet-ON system resulted in sequential increases of the Prrx1 and TNC protein (**e**) and mRNA (**f**) expression in normal lung fibroblasts (MRC5). **g** Schematic diagram of the Twist1-Prrx1-TNC PFL

(Fig. 2f). A homology study also showed that there is strong consensus in the sequence of this region among 11 vertebrates (Supplementary Fig. 2). These data strongly suggest that Twist1 is a potent positive regulator of Prrx1 in fibroblasts (Fig. 2g).

**Twist1, Prrx1, and TNC form a PFL in fibroblasts.** Next, we examined whether Prrx1 directly upregulates TNC expression in fibroblasts, and we found that the exogenous expression of both splicing isoforms of Prrx1 (Prrx1a and Prrx1b) enhanced both mRNA and protein levels of TNC in normal fibroblasts (Fig. 3a and Supplementary Fig. 3a); in contrast, shRNA-mediated knockdown of Prrx1 led to downregulation of both mRNA and protein levels of TNC (Fig. 3a and Supplementary Fig. 3a). In addition, transient Prrx1 expression also enhanced the transcriptional activity of the TNC promoter in fibroblasts (Fig. 3b). These results are highly consistent with previous studies that showed TNC regulation by Prrx1 in vascular smooth muscle[22] and fibroblasts[23]. Next, we found that Prrx1 is indispensable for Twist1-induced TNC expression. In normal fibroblasts, Twist1 failed to induce the expression of both protein and mRNA levels of TNC levels in the absence of Prrx1 (Fig. 3c, d).

If Twist1, Prrx1, and TNC formed a PFL, Prrx1 and TNC genes were predicted to be sequentially activated upon Twist1

induction. As expected, in response to the doxycycline-induced Twist1 expression, Prrx1 expression was induced and then sequentially followed by TNC; induction of TNC expression was delayed for 8 days (MRC5) and 3–5 days (IMR90 and SNF#19) after doxycycline treatment, as shown in Fig. 3e, f, and Supplementary Fig. 3b, and these results confirm the presence of a PFL that comprises Twist1, Prrx1, and TNC (Fig. 3g).

**Systems biology study reveals the role of Twist1-Prrx1-TNC PFL.** Positive feedback regulation can amplify a signal, induce an ultrasensitive (all-or-none) response to a stimulus, or create a bistable switch[24,25]. To investigate the significance and functional implications of the Twist1-Prrx1-TNC PFL, we developed a mathematical model (Fig. 4a) based on the gene expression kinetics of each component of the PFL in MRC5 fibroblast cell line in response to doxycycline-mediated Twist1 induction (Supplementary Fig. 4a), and we explored how the positive feedback regulation shaped the response profiles of individual components to the varying doxycycline stimuli. The PFL can produce ultrasensitivity, which enables the components to respond to the doxycycline stimulus in an all-or-none manner. When doxycycline stimulation increases, the dynamical state of the system remains at the lower branch of the stimulus-response

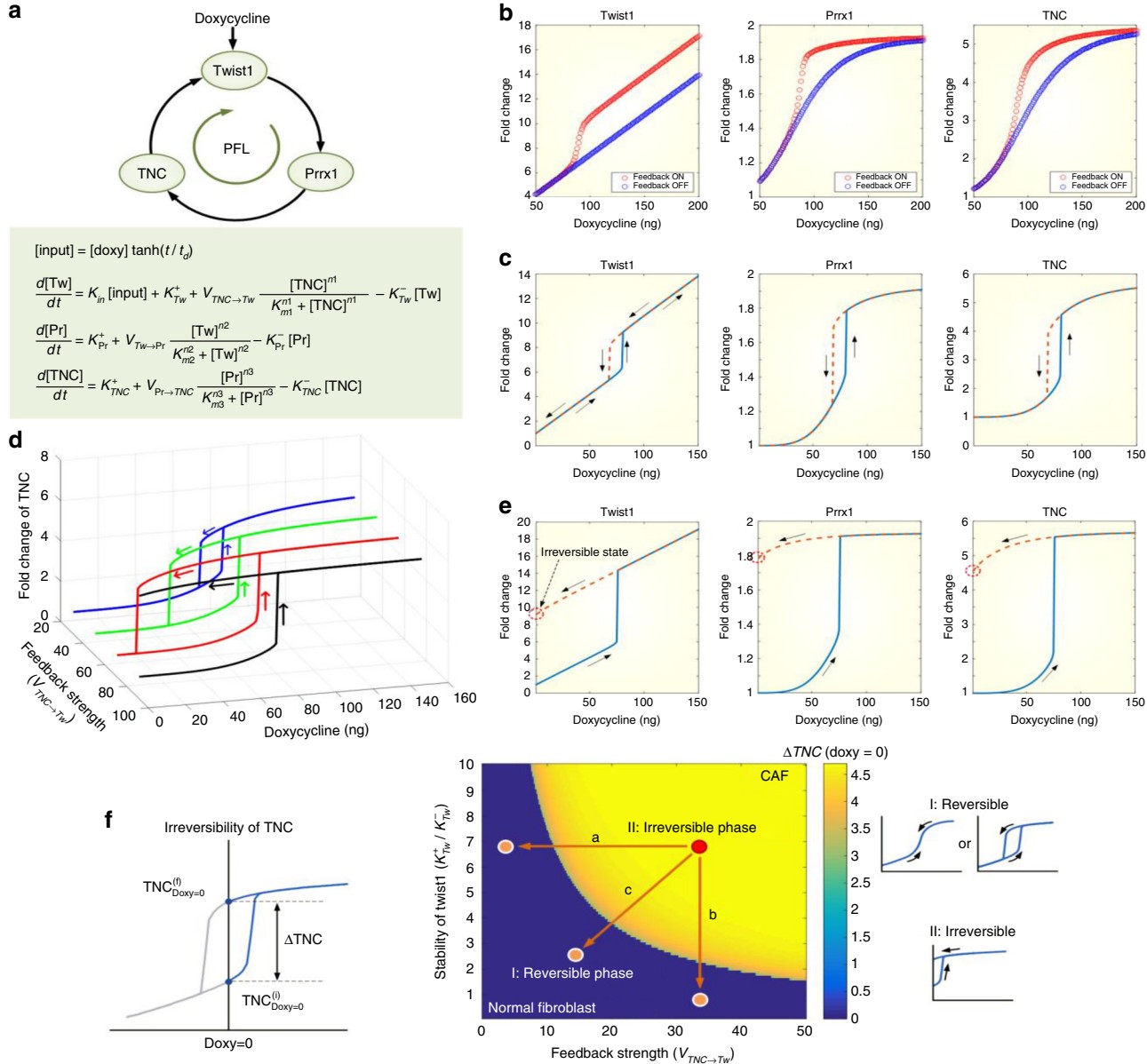

**Fig. 4** The Twist1-Prrx1-TNC PFL functions as an internal bistable switch in regulating the proliferative signal of fibroblasts. The importance and functional implications of the Twist1-Prrx1-TNC PFL were demonstrated through mathematical simulations based on experimental data. **a** Schematic diagram of the Twist1-Prrx1-TNC PFL and the mathematical description of the loop (see Supplementary text for detailed explanations). **b** Ultrasensitive responses of the PFL to the varying doxycycline stimuli compared with cases in which feedback regulation was absent ($V_{TNC \to Tw} = 0$). **c** Hysteretic responses of the three genes to the doxycycline stimulus. **d** Various profiles for the hysteretic TNC response to the doxycycline stimulation along with the increase of the feedback strength ($V_{TNC \to Tw}$). **e** Irreversible hysteretic responses of the three genes to the doxycycline stimulus when the feedback strength is large ($V_{TNC \to Tw} = 80$). **f** Phase diagram illustrating the irreversibility of TNC with respect to the feedback strength ($V_{TNC \to Tw}$) and stability of Twist1 ($K_{Tw}^{+}/K_{Tw}^{-}$). $\Delta TNC$ denotes $TNC^{fin} - TNC^{ini}$ when doxycycline = 0 (left). Reversibility was characterized by TNC = 0 (irreversible, otherwise). The irreversibility of TNC can be categorized into two phases (center). Representative hysteretic curves for the corresponding phases are shown in the right panel

curve, and the expression level of the three genes increases slowly, until the input reaches a certain threshold (Fig. 4b). Beyond the threshold, the system state switches abruptly to the other branch, where the genes' expression levels are constitutively high; this ultrasensitivity is accompanied by a hysteretic response to doxycycline stimulation. The doxycycline concentration should reach a critical threshold to activate the three genes from their basal states; however, once they are activated, each gene can maintain its active state even after the input decreases below the threshold (Fig. 4c). In combining this information, we conclude that the Twist1-Prrx1-TNC PFL forms an internal bistable switch in regulating the proliferative fibroblast signal.

To further explore the role of hysteretic bistable switching in fibroblast proliferation under pathological conditions, we examined the hysteretic response profiles of TNC along with the increase of Twist1. Increasing the production rate, $K_{Twist1}^{+}$, or decreasing the degradation rate, $K_{Twist1}^{-}$, of Twist1 promotes the accumulation of TNC, which results in lowering the critical threshold of doxycycline required to activate TNC (Supplementary Fig. 4b, c). In contrast, increasing the production rate, $V_{TNC \to Twist1}$, of Twist1 by TNC, maintained the critical threshold of the stimulus but enlarged the hysteresis width. This indicates that sustained Twist1 activation by highly activated positive feedback regulation is essential for robust bistable switching in

regulating fibroblast proliferation (Fig. 4d). Intriguingly, we also found that attaining sufficient Twist1 production by TNC can make the TNC switching process irreversible. In other words, the levels of Twist1, Prrx1, and TNC do not completely return to their initial states but rather remain in their active states even after the doxycycline stimulus is removed (Fig. 4e).

To quantitatively analyze the role of Twist1 in determining the irreversible hysteretic response, we defined a measure for the irreversibility of TNC, $\Delta TNC$. By analyzing the irreversibility with respect to the stability of Twist1 (the ratio of the production rate to the degradation rate, $K_{Tw}^+/K_{Tw}^-$) and the feedback strength (the TNC-induced production rate of Twist1, $V_{TNC \to Tw}$), we found that TNC activation can be categorized into two phases: reversible or irreversible (Fig. 4f). $\Delta TNC = 0$ in Phase I indicates a reversible state of TNC, whereas $\Delta TNC > 0$ in Phase II denotes an irreversible state. A pathologic state, such as CAF, can be represented by an irreversible proliferative state (Phase II) in which Twist1 is highly activated, and thus, the number of fibroblasts might increase drastically. Taken together, we suggest that the Twist1-Prrx1-TNC PFL functions as an internal bistable switch in regulating fibroblasts activation and that the persistent activation of Twist1 caused by the PFL might contribute to the perpetual fibroblast activation under pathologic conditions.

**The PFL generates bistability in fibroblast activation.** According to our computational analysis, Twist1-Prrx1-TNC PFL was predicted to function as a bistable switch, allowing only a very low basal level (ground state) or a highly expressional level (active state) of the three genes. To validate this prediction, the impact of Twist1 on the expressional status of other genes was examined in MRC5 fibroblasts using Doxycycline-mediated Twist1-induction system and multiplex immunohistochemistry (mIHC). mIHC allows simultaneous detection of all three genes in individual MRC5 fibroblasts. Highly consistent with computational prediction, the expressions of all three of these genes followed a bimodal distribution, ON or OFF, a hallmark of a bistable system (Fig. 5a).

The same pattern of bistable triple gene activation was also observed in fibroblasts during cutaneous wound healing (Fig. 5b). After circle-full-thickness cutaneous wounds (diameter of 5 mm) were made on the dorsal skin of mice, the expressions of three components of the Twist1-Prrx1-TNC PFL in fibroblasts was determined, at multiple time points, using mIHC. All three genes were not expressed at day 0, but Twist1 began to be abruptly expressed in fibroblasts from day 1 after the wounds had been created, subsequently followed by Prrx1 and TNC expression. Interestingly, the periods in which these three genes were intensively coexpressed (days 5~9) completely overlapped with the period of maximum fibroblast proliferation. In the middle and late phases of wound healing, the levels of these three genes fell sharply, and the number of fibroblasts rapidly decreased. Twist1 and TNC levels were maintained for a relatively longer, possibly due to the hysteretic response of the PFL, as predicted in Fig. 4d, e (Fig. 5c). These three gene expression patterns perfectly matched the kinetics of fibroblasts during wound healing, highlighting the significance of the Twist1-Prrx1-TNC PFL in fibroblast activation. Interestingly, the bimodal distribution of the three genes' expression was observed again, which is in good agreement with the computational prediction (Fig. 5d).

Furthermore, we found that the bimodal expression of these three genes was directly correlated with fibroblast activation in CAF. The GFP reporter, under the control of a Twist1 promoter, was stably integrated into the chromosome of two patient-derived CAFs (Fig. 5e and Supplementary Fig. 5). Then, the CAFs were sorted into Twist1-positive and Twist1-negative by fluorescence-activated cell sorting (FACS) (Fig. 5e). Interestingly, the Twist1-

positive CAFs were also positive for both Prrx1 and TNC, whereas the Twist1-negative CAFs were negative for both, showing a bimodal distribution of the triple gene expression (Fig. 5f and Supplementary Fig. 5). The Twist1-positive CAFs also demonstrated both significantly enhanced proliferation capacity (Fig. 5f and Supplementary Fig. 5) and a resistance to apoptosis induced by doxorubicin (Fig. 5g). These results suggest that the Twist1-Prrx1-TNC PFL indeed generates bistability in fibroblast activation.

**Sustained switching ON of the PFL triggers fibrotic nodules.** Next, we attempted to assess the in vivo significance of the Twist-Prrx1-TNC PFL as a bistable (ON/OFF) switch. For this, we induced the sustained "Switch ON" condition of this PFL in vivo by treating transiently activated fibroblasts with exogenous TNC in the mouse wound-healing model because our mathematical model predicted that TNC is a key element for irreversible Twist1-Prrx1-TNC PFL activation. TNC has recently been reported to strongly induce fibrosis in various organs by creating a fibrogenic niche[26] and damage-associated molecular pattern[27]. Thus, we examined whether this sustained PFL activation would trigger persistent fibroblast activation in vivo. To rule out another possibility that the effect of exogenous TNC is mediated by a mechanism other than Twist1-Prrx1-TNC PFL, we repeated the same in vivo experiment in the absence of PFL. Specifically, we injected single high dose of TNC (10 μg) into the wound site 3~5 days after skin excision in wild type mice and we performed the same experiment again in fibroblast-specific Twist1 knockout mice in which Twist1-Prrx1-TNC PFL was disrupted (Fig. 6a, b).

All of these mice were necropsied in the remodeling phase of wound healing and thoroughly examined by pathologists, and in the wild-type mice (Fig. 6c iii. WT mice), exogenous TNC generated multifocal fibroblastic nodules that were histologically similar to the fibrotic foci of IPF whereas in the wound sites of the control group (mice injected with PBS), the fibroblasts exhibited a nonproliferative and mature fibroblast phenotype, which is highly compatible with the normal remodeling phase (Fig. 6c). This fibroblastic nodule is mainly composed of hyper-proliferative immature fibroblasts and epithelial hyperplasia, and these findings are also a hallmark of IPF (Fig. 6c and Supplementary Fig. 6a). Remarkably, this fibroblastic nodule persisted beyond the remodeling phase (experiments were conducted three times: first trial, day 14 after skin excision; second trial, day 25, and third trial, day 20). However, we did not observe these exogenous TNC-induced phenotypes at all in the fibroblast-specific Twist1 knockout mice. In addition, mIHC revealed that fibroblasts expressing all three genes, Twist1, Prrx1, and TNC, were exclusively concentrated within fibrotic nodules generated by exogenous TNC (Fig. 6c, bottom). All these findings confirm that exogenous TNC triggered irreversible fibroblast activation through the perpetual switching on of the Twist1-Prrx1-TNC PFL.

A closer look at this in vivo fibrotic nodule revealed that Twist1-highly positive fibroblasts were observed only in certain area near epithelium (Dermis zone1) where TNC was very highly enriched (Fig. 7a and Supplementary Fig. 7). This interesting finding is highly consistent with our mathematical model (Fig. 4) and suggests that very high TNC levels may be required to permanently activate the PFL in vivo. To understand the detailed mechanism underlying this in vivo phenomenon, we performed additional computational analysis based on mathematical modeling of the experimental data. For the precise mathematical modeling, we obtained quantitative experimental data by examining the responses of ex vivo cultured mouse dermal

fibroblasts treated with various concentrations of TNC (0, 0.1, 1, 2, and 5 μg). The result was that the mouse dermal fibroblasts started to express significant endogenous TNC and Twist1 only when treated with over 5 μg of exogenous TNC (Fig. 7b). This result strongly suggests that the PFL is activated only above a

certain threshold, again highlighting the significance of TNC in PFL activation.

Using these western blot data, we constructed the mathematical model by modifying the Twist1-Prrx1-TNC PFL such that the key parameter values related to Twist1 and TNC levels were

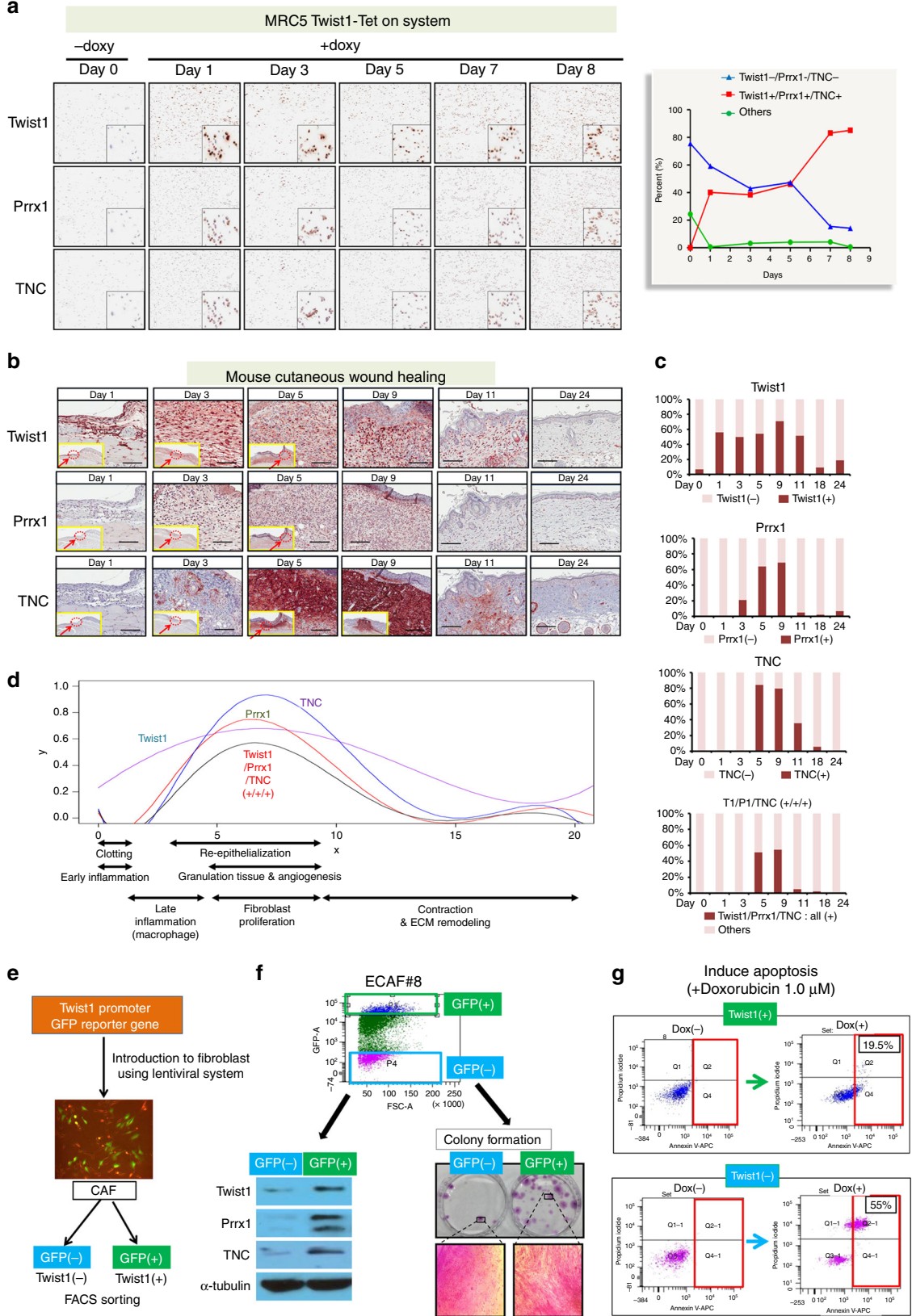

in accordance with the experimental results shown in Fig. 7b (see Supplementary text for further details on the mathematical model). Both our experimental and computational results show that the TNC injection induced an abrupt elevation of Twist1 and TNC in a steep and nonlinear manner. Such a sigmoidal dose−response behavior, termed ultrasensitivity, has been known to be an important feature of positive feedback. To further investigate the role of the Twist1-Prrx1-TNC PFL in persistent TNC activation upon injection of a high TNC level, we explored the difference of TNC level between initial and terminal state after removal of initial stimulus, represented by $\Delta TNC$, with respect to the feedback strength ($V_{TNC->TW}$) and the TNC level required for feedback activation ($K_{m1}$; Fig. 7c, top left). When $V_{TNC->TW}$ is high and $K_{m1}$ is low, the irreversibility of the PFL is exhibited ($\Delta TNC>0$). In the irreversible state, once the PFL is activated by TNC injection, TNC level does not return to initial state even if initial stimulus disappears due to hysteretic response of the PFL (Fig. 7c, top right). The TNC level in the reversible state gradually changes along with the increase or decrease of the stimulus (Fig. 7c, bottom left). And in the reversible state, the TNC level returns to its initial state ($\Delta TNC=0$) as the stimulus decreases back to zero regardless of the level of the recombinant TNC concentration, $TNC^{max}$ (Fig. 7c, bottom left and right). However, the TNC level in the irreversible state does not completely return to its initial state and just remains at a certain active state ($\Delta TNC>0$) in response to initial stimulus with a high level of recombinant TNC above a certain threshold (e.g., $TNC^{max} = 3$) (Fig. 7c, bottom right).

Therefore, this experimental data based computational analysis strongly suggests that the Twist1-Prrx1-TNC PFL can be irreversibly activated in response to a certain high level of TNC injection. This is well in accordance with our *in vivo* findings in which fibroblasts expressing high levels of TNC, Twist1, and Prrx1 were found only in fibrotic nodules. Our combined *in vivo* and computational studies raised the possibility that high level of TNC could induce some pathologic fibrosis by irreversibly activating the Twist1-Prrx1-TNC PFL of fibroblasts.

Recently, TNC has been strongly emphasized as a potent inducer of fibrosis as well as a therapeutic target of fibrotic diseases[50–52]. However, there is a missing link regarding detailed mechanisms for how TNC has such a crucial impact on the decision-making process of fibroblasts to initiate fibrosis. The findings from this study suggest that the Twist1-Prrx1-TNC PFL may be that link, functioning as a switch triggered by TNC to start fibrosis.

**The clinical implications of the Twist1-Prrx1-TNC PFL.** Based on our in vitro, in vivo and computational studies strongly suggesting that Twist-Prrx1-TNC PFL may induce perpetual fibroblast activation under pathologic conditions, we further investigated clinical implications of this PFL in CAFs and fibrotic diseases. First, we examined coexpression of key component of the PFL; Twist1, Prrx1, and TNC in clinical samples such as patient tissue-derived ex vivo cultured CAFs, archived paraffin-embedded cancer tissues, and TCGA data. First of all, the concurrent coexpression of Twist1, Prrx1, and TNC was prevalent in ex vivo cultured CAFs (Supplementary Fig. 8a). Analysis of RNA-SEQ data downloaded from TCGA revealed a very strong correlation among the mRNA levels for all three genes in various types of cancers (Supplementary Fig. 8b). Furthermore, mIHC revealed that these three genes were intensively coexpressed only in CAFs within patients' cancer tissues (Fig. 8a). These data strongly indicate that the coexpression of these three genes in CAFs is common in various types of cancers.

To further investigate the biological functions that are related to the Twist1-Prrx1-TNC PFL in cancer, we analyzed the mRNA expression data of 11 types of cancers downloaded from TCGA using a gene set enrichment assay (GSEA). The results show that gene signatures related to ECM remodeling, angiogenesis, apoptosis, epithelial−mesenchymal transition, and p53 suppression were significantly enriched in the Twist1/Prrx1/TNC (+/+/+) group in all 11 cancer types (Supplementary Fig. 8c and Supplementary Table 3). These findings are highly consistent with previous studies reporting on Twist1's role in fibroblast activation[13,16,28–30].

Next, we investigated the clinical relevance of CAFs that exhibit the Twist1-Prrx1-TNC PFL. To this end, we examined the detailed expressional profiles of these three genes in individual CAFs of 306 esophageal cancer cases using mIHC. We found that CAFs concurrently expressed all three genes in 38.9% of the cases (119/306; Fig. 8b). Interestingly, we also found that in 39.5% of the cases (121/306), CAFs expressed none of these genes. Therefore the bimodal distribution of the triple gene expression, a hallmark of a bistable system, was evident again. There was also a very strong association between the Twist1-Prrx1-TNC PFL expression in CAFs and the patients' poor prognosis in esophageal cancer ($P < 0.001$; Fig. 8c).

Finally, we studied the expressions of these genes in individual fibroblasts within the paraffin-embedded patients' tissues of various fibrotic diseases including IPF, keloid disease, and desmoid tumor (aggressive fibromatosis) using mIHC. The results showed that all three genes tended to be frequently coexpressed only in pathologic fibroblasts, especially in the "fibrotic hot spot" of all these fibrotic diseases and cancer stroma (Fig. 8d and Supplementary Figs. 8d, e and f).

These findings strongly suggest that a PFL comprised of Twist1, Prrx1, and TNC is highly likely to play an important role in pathologic fibroblasts in cancer stroma and fibrotic diseases (Fig. 9).

---

**Fig. 5** The role of the Twist1-Prrx1-TNC PFL as a bistable switch was validated using an in vitro fibroblast line, cutaneous wound healing, and patient-derived CAFs. **a** Twist1 expression was induced in fibroblasts (MRC5) using a doxycycline-mediated induction system, and we examined the impact of the induced Twist1 on the Prrx1 and TNC expression profiles using multiplex immunohistochemistry (mIHC). Bimodal distribution of the triple gene expression, such as all positive vs. all negative, was evident after 7 days of induction. The insert images are enlarged picture of stained cells in each staining images. **b** We examined the Twist1, Prrx1, and TNC expressions during cutaneous wound healing using multiplex immunohistochemistry. The yellow box is a low power view of the mouse wound area. In this yellow box, dashed circle and arrow indicate area of interest (AOI) which is shown as a high power view. **c** Quantitative analysis of Twist1, Prrx1, and TNC expression in fibroblasts during wound healing. After we made circle-full-thickness cutaneous wounds (diameter of 5 mm) on the dorsal skin of mice, we determined the Twist1, Prrx1, and TNC expression in fibroblasts at multiple time points using mIHC. We noted highly dynamic changes in these three genes' expression in fibroblasts. **d** The plot is generated based on data derived from Fig. 5b. Fibroblasts that were positive for all three genes were restricted to the granulation tissue formation stage of wound healing. **e** The GFP reporter, under the control of a Twist1 promoter, was stably integrated into the chromosome of two patient-derived CAFs. Then, we sorted CAFs into Twist1-positive and Twist1-negative CAFs, using flow cytometry. **f** Esophageal cancer-derived CAFs (ECAF#8) were sorted into Twist1-positive and Twist1-negative CAFs. Twist1-positive CAFs were specifically positive for Prrx1 and TNC and also exhibited increased colony-forming abilities. **g** The Twist1-high CAF (ECAF#8) showed resistance to doxorubicin-induced apoptosis

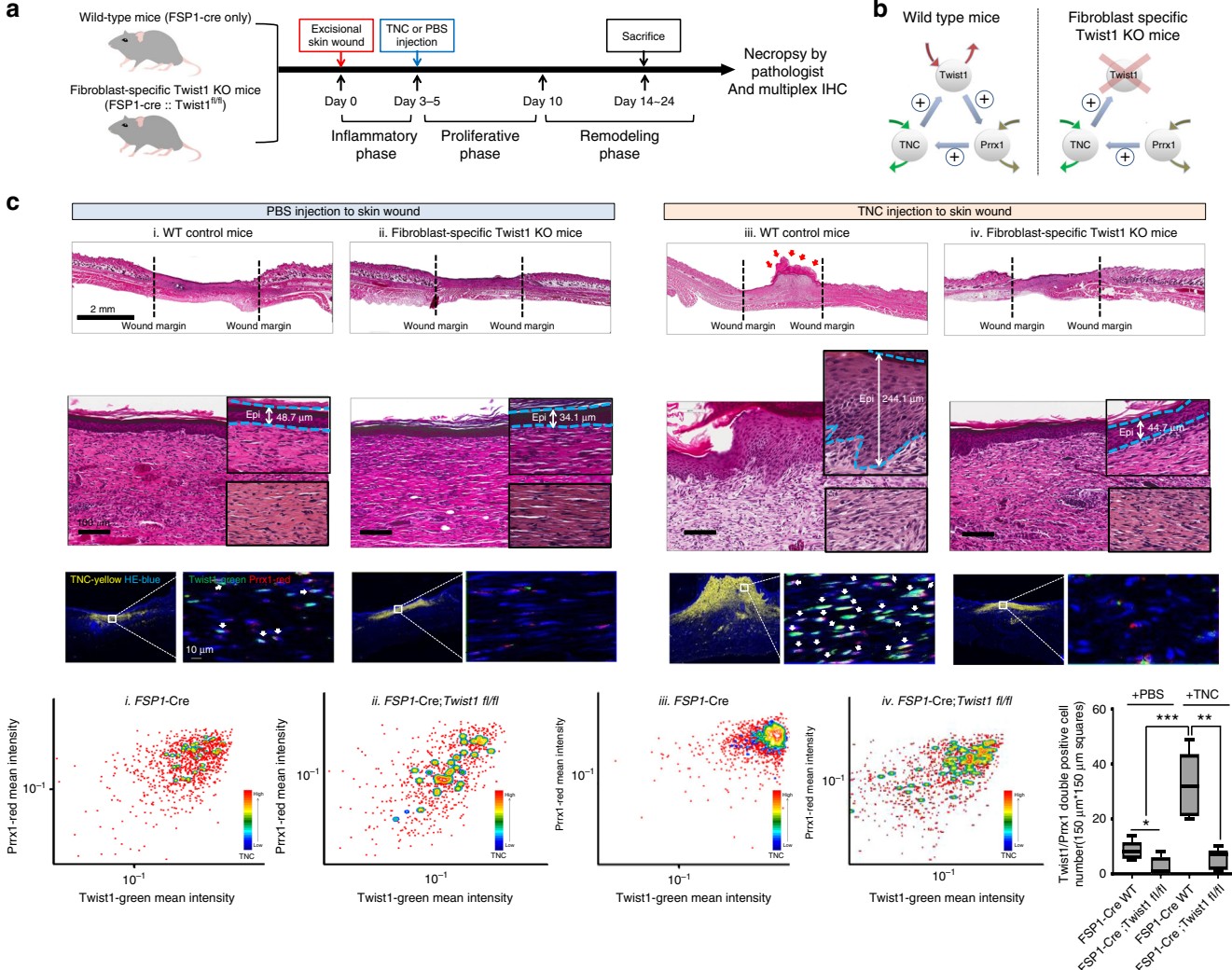

**Fig. 6** Sustained switching ON of the Twist1-Prrx1-TNC PFL triggers persistent fibroblast activation, forming fibrotic nodules in vivo during wound healing. **a** Schematic diagram of the experimental design. We injected TNC protein or PBS (control) into the wound area of WT and fibroblast-specific Twist1 knockout mice. The experiments were conducted three times ($N = 5$ mice in each experimental group). Experimental timeline for the detailed procedures is indicated. **b** The maintenance of the PFL activation by TNC injection (FSP1-cre; Twist1 WT mice, left) and disruption of PFL by fibroblast-specific Twist1 deficiency despite the TNC injection (FSP1- Cre;Twist1$^{fl/fl}$ mice, right). **c** H&E staining images showing the wound areas of each group (top). In Twist1 WT mice (depicted as iii, FSP1-cre; Twist1 WT), we observed fibroblastic nodules (red arrow) comprising hyper-proliferative immature fibroblasts and epithelial cell hyperplasia (blue dotted line). TNC is strongly expressed in most fibroblastic nodules as mentioned above in which Twist1 and Prrx1 are also strongly coexpressed. These results were obtained through multiplex IHC staining and visualized using ImageJ Fiji (Twist1-green, Prrx1-red, TNC-yellow, and hematoxylin-blue). We performed quantitative analysis of these results using FCS Express 6 image cytometry analysis software, and we presented the outcomes as an FACS plot (x axis-Twist1 green, y axis-Prrx1 red and contour indicated with TNC) (bottom). The number of fibroblasts displaying Twist1/ Prrx1 double positivity in the TNC- or PBS-injected wound areas (150 μm × 150 μm square) is indicated by box plot (bottom right of Fig. 6c). Data are presented as the mean ± SEM; $N = 6$ independent measurements (two-tailed $t$ test: **$p<0.005$, ***$p<0.001$)

## Discussion

Fibroblast activation is a very critical event in wound healing, chronic fibrotic diseases, and cancer. In particular, the functional changes observed during the activation of quiescent fibroblasts in wound healing is so dramatic that fibroblast activation is comparable with fundamental cell fate change[31,32]. For fibroblast to correctly initiate such an important activation process at the right time, it requires complex logic circuit that determines fibroblast activation by integrating and processing various input information. Although several factors responsible for fibroblast activation have been reported, the complex logic circuit has never been studied. In search of this circuit, we investigated the key transcription factors that induce fibroblast activation because transcription factors often make fundamental decisions for cell fate

change, known as transcription factor-mediated reprogramming[33]. In our previous study, we found that Twist1 was a key transcription factor for fibroblast activation, and we also found that Twist1 strongly enhanced TNC expression[13,34,35]. Recent studies have indicated that TNC is a critical determinant of fibroblast activation by providing either key information such as damage-associated molecular pattern[27] or fibrogenic niche[26]. Therefore, we speculated that the Twist1-TNC axis might be a part of the complex logic circuit. In this study, while studying Twist1-TNC axis, we accidentally found the positive feedback loop comprising Twist1, Prrx1, and TNC. Mathematical, clinical, and in vivo studies indicated that this PFL functions as an "ON/ OFF switch", a key unit of a complex logic circuit. Most interestingly, this Twist1-Prrx1-TNC PFL can be permanently

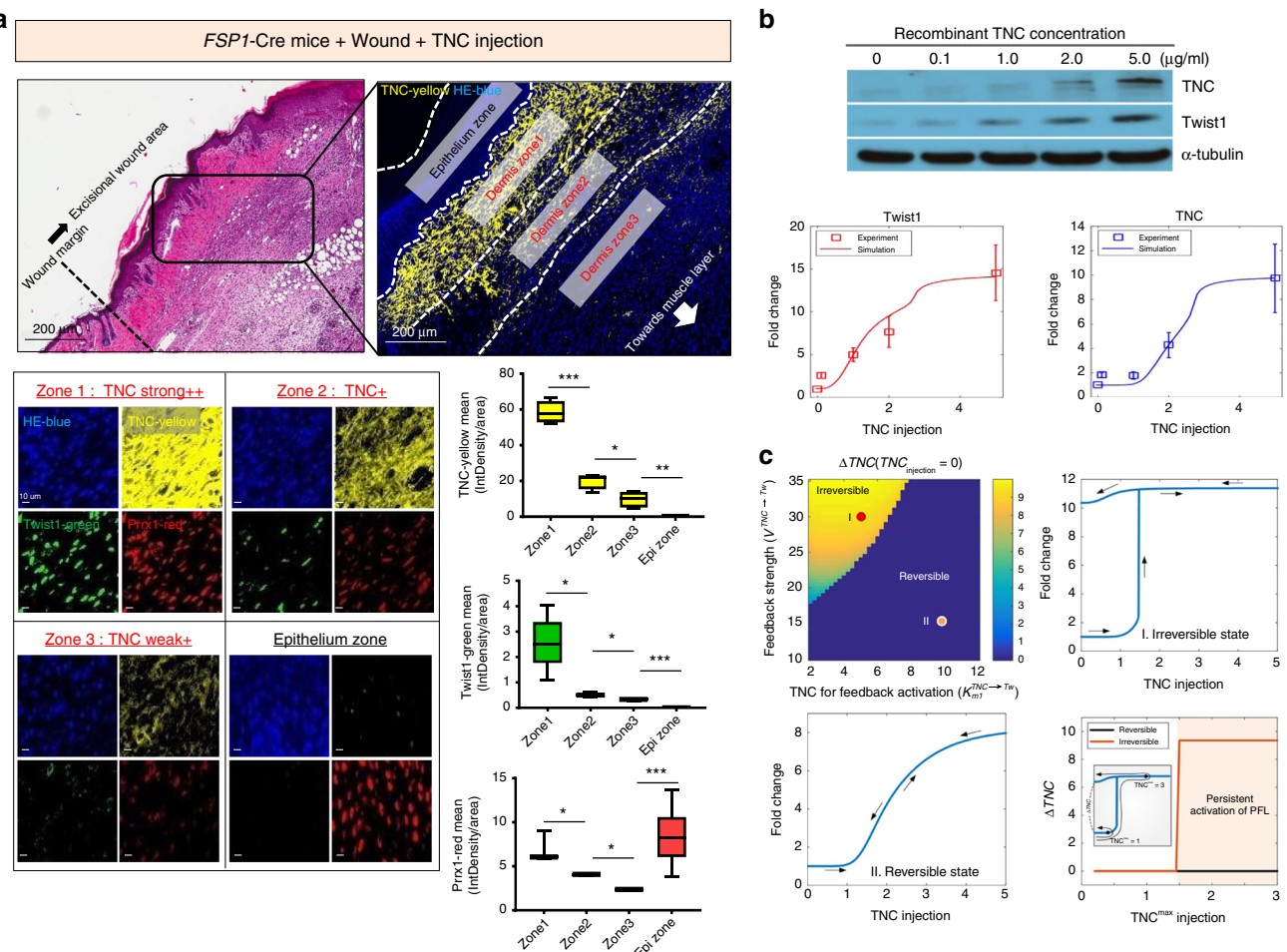

**Fig. 7** High TNC is essential for activating the irreversible PFL. **a** In the fibrotic nodule generated by exogenous TNC, we observed a TNC concentration gradient (top). In-depth analysis of this nodule using multiplex IHC showed that Twist1 and Prrx1 were strongly positive and coexpressed in most fibroblasts within strongly TNC-positive zone1, and this pattern disappeared as the TNC concentration decreased (bottom left) (Twist1-green, Prrx1-red, TNC-yellow and hematoxylin-blue). We measured each integrated density per zone representing the expression level of each gene using ImageJ Fiji (bottom right). Data are presented as the mean ± SEM; $N = 3$ independent measurements (two-tailed $t$ test: $*p < 0.05$, $**p < 0.001$, $***p < 0.0001$). **b** We treated mouse dermal fibroblasts isolated from fresh Balb/c mouse tissue with TNC at different concentrations (0, 0.1, 1, 2, and 5 μg) and then evaluated Twist1 and TNC expression by western blotting (top). We estimated the kinetic parameter values of the mathematical model based on this western blotting data on the Twist1 and TNC fold changes along with the increase of exogenous TNC (bottom). Discrete data marked with squares indicate experimental measurements of the corresponding genes ($N = 3$, error bars indicate s.d.). **c** Phase diagram illustrating the irreversibility of TNC (represented by color intensity) with respect to the feedback strength (V) and the TNC level for feedback activation ($K_{m1}$, the Michaelis−Menten constant) (top left). The hysteretic responses of TNC to the TNC injection in the irreversible (($K_{m1}$, V) = (5, 30)) and reversible (($K_{m1}$, V) = (10, 15)) states are shown in the top right and bottom left figures, respectively. Given that the stimulus increases from zero to $TNC_{max}$ and then decreases back to zero, we compared the irreversibility of TNC, ΔTNC, with that of the reversible state (bottom right). The inlet shows examples of obtaining ΔTNC for the $TNC_{max}$

switched ON, leading to irreversible activation of fibroblasts under pathologic conditions.

The importance of each member of this PFL has been reported by previous studies[13,22,30,36]. The significance of Twist1 in fibroblast activation was highlighted by various in vitro, clinical, and in vivo studies including our previous report. In particular, Twsit1 is strongly implicated in cancer progression and various fibrotic diseases such as systemic sclerosis, IPF, desmoid tumors, and keloid diseases[16,29,30,37]. Recently, TNC has emerged as an essential determinant of fibroblast activation; for instance, it was revealed to induce persistent fibroblast activation by functioning as a damage-associated molecular pattern[27]; TNC was also shown to generate the fibrogenic niche in kidney fibrosis[26]. One interesting point is that all three genes are highly expressed in embryo-mesenchymal cells, precursors of fibroblasts. In contrast, in adult

tissue, all three genes are expressed only in wound healing, cancer, and fibrotic tissue. Furthermore, both Twist1 and Prrx1 knockout mice are embryonically lethal due to the failure of mesenchymal cell activation[22,38]. All of these studies show that this PFL is composed of very important and essential factors for fibroblast activation, which highlights its significance and reliability.

In this study, Twist1- Prrx1-TNC PFL was revealed to be functioning as a bistable (ON/OFF) switch to initiate fibroblast activation. Recently in vivo circuits functioning as switches have been known to be abundant and important for decision-making in various biological functions such as cell cycles, differentiation, and apoptosis progression[8,9]. Blaauboer et al.[39] anticipated that some sort of PFL is necessary for fibroblast activation; in addition, they speculated that TNC may be a key member of a possible PFL

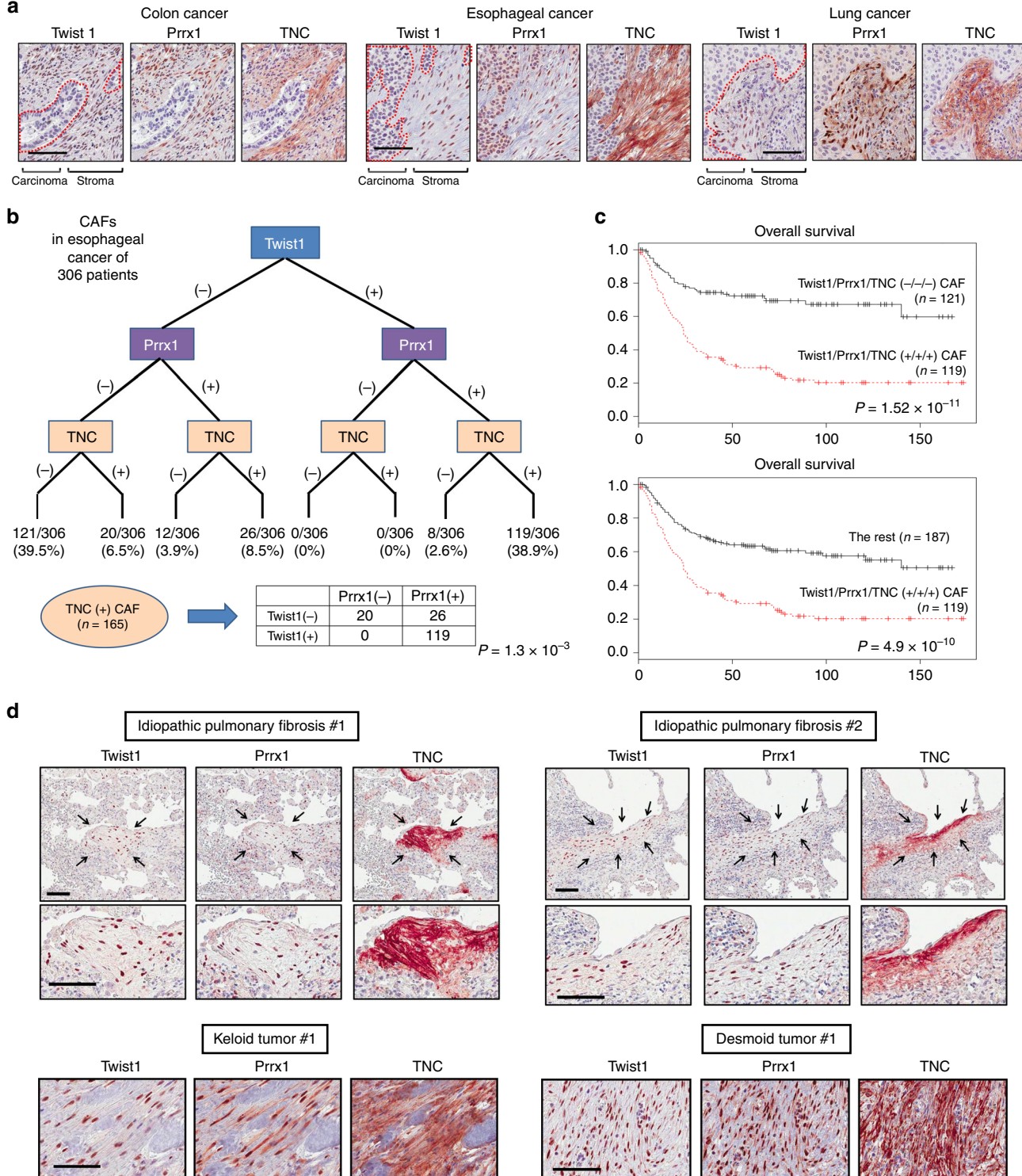

**Fig. 8** The coexpression of Twist1, Prrx1, and TNC is frequently found in cancer-associated fibroblasts and pathologic fibrosis. The clinical implications of the Twist1-Prrx1-TNC PFL in CAFs and fibrotic diseases. **a**, **b** We examined the coexpressions of all three genes by mIHC (scale bar indicates 100 μm). The bimodal distribution of the triple gene expression, all positive vs. all negative, a hallmark of a bistable system, was evident again in esophageal CAFs. These data are highly consistent with the computational prediction that the Twist1-Prrx1-TNC PFL functions may lead to irreversible and perpetual activation in pathologic conditions. We analyzed the expression profiles of these three genes in 306 esophageal cancer tissues. **c** The presence of CAFs that were positive for all three genes was strongly associated with the worse clinical outcomes. **d** Fibroblasts that were positive for all three genes were frequently observed in pathologic fibrosis, such as idiopathic pulmonary fibrosis, keloid disease, and desmoid tumors (aggressive fibromatosis). Scale bar indicates 100 μm

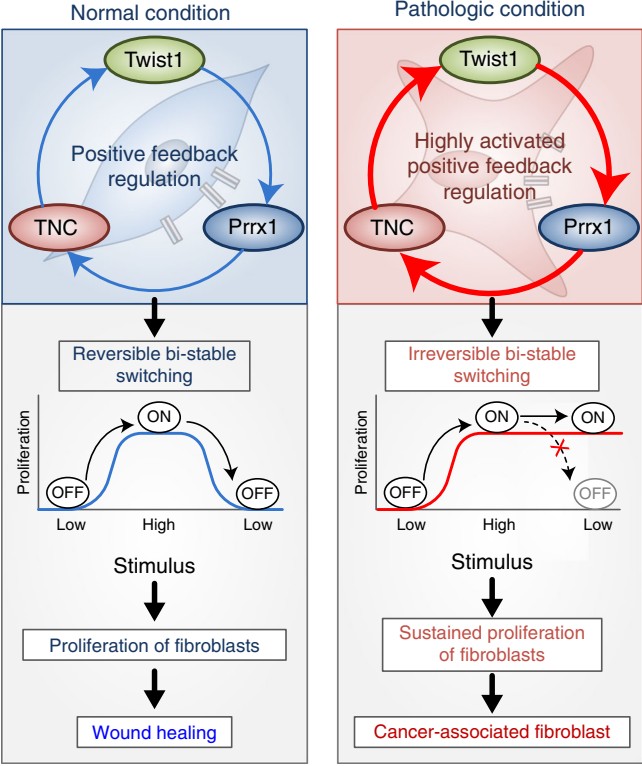

**Fig. 9** Summary of our study. The Twist1-Prrx1-TNC positive feedback regulation provides clues for understanding the activation of fibroblasts during wound healing under normal conditions, as well as abnormally activated fibroblasts in pathological conditions such as cancerous and fibrotic diseases. Under normal conditions, the PFL acts as a reversible bistable switch by which the activation of fibroblasts is "ON" above a sufficient level of stimulation and "OFF" for the withdrawal of the stimulus. However, this switch can be permanently turned on under pathologic conditions by continued activation of the PFL, resulting in sustained proliferation of fibroblasts

without any specific experimental data. In this study, we identified the complete members of the PFL that activates fibroblasts with extensive experimental evidences including in vivo and clinical studies. In wild-type mice, the PFL strongly induced fibroblast activation in vivo, whereas in fibroblast-specific Twist1 knockout mice, this loop failed to be initiated. The expression pattern of all three genes in these mice models was highly consistent with the PFL-based mathematical prediction. In patient-derived cancer and fibrotic clinical tissues, all three genes were specifically highly expressed in the activated fibroblasts with the bimodal distributional pattern, the hallmark of a PFL (Fig. 8b). Another line of evidences was from a series of meticulous in vitro studies. Each step of the PFL was thoroughly verified using various ex vivo fibroblast cultures and extensive biochemical experiments.

Our systems biology approach integrating computational and in vivo studies revealed the detailed mechanism of how Twist1-Prrx1-TNC PFL functions as a bistable (ON/OFF) switch. In addition, one very interesting possibility was predicted from our systems biology analysis: This switch can be permanently turned on under pathologic conditions. Specifically, our mathematical model predicted that this PFL could be irreversibly activated when the strength of the TNC->Twist1 feedback was above a certain threshold. In this situation, the hysteresis of this PFL is extremely exaggerated and it never returns to the initial state. We hypothesized that persistently switching on the Twist1-Prrx1-

TNC PFL would lead to perpetual activation of fibroblasts, ultimately resulting in fibrotic diseases. We confirmed this hypothesis in our in vivo study in which persistent PFL activation was reproduced by treating wounded tissue with exogenous TNC. Indeed, exogenous TNC-mediated persistent PFL activation not only led to hyperactivation of fibroblasts but also generated fibrotic foci very similar to IPF.

Initially, we intended to find a complex logic circuit that would function as a decision maker for fibroblast activation. To process multiple layers of input information, such as ECM, mechanical tension, various growth signals, and a myriad of internal signals, the complex logic circuit is assumed to be composed of numerous switches and units that perform logical operations. The Twist1-Prrx1-TNC PFL is presumed to be a subsystem of this complex logic circuit. Even though we did not reveal the entire complex logic circuit, our study provides a solid and concrete basis for discovering the whole picture.

Another interesting prediction we inferred from the Twist1-Prrx1-TNC PFL is the presence of a negative regulator of this PFL. Basically, this negative regulator is supposed to be another part of the complex logic circuit. During normal wound healing, the Twist1-Prrx1-TNC PFL is turned ON in the fibroblast proliferative phase and then turned OFF in the late remodeling phase. For the PFL to be switched OFF at the appropriate time, the negative regulator should respond to multiple layers of input information. Fibroblast activation should be terminated at the appropriate time only after epithelial closure, adequate mechanical tension, and sufficient ECM. In particular, the machinery-degrading ECM including matrix metalloproteinases (MMPs) is suggested to be a critical component of the negative regulator of this PFL according to our computational analysis (Supplementary Fig. 9)[40–42]. In accordance with this hypothesis based on computational analysis, several MMP-deficient mice were documented as displaying a persistently activated fibroblast phenotype associated with fibrotic diseases in multiple organs[43–45]. Therefore, the next challenge will be to uncover this negative regulator, which is crucial in maintaining the homeostasis of fibroblasts during wound healing. Overall, our model can be a starting point for exploring this negative regulator.

Because the Twist1-Prrx1-TNC PFL plays a central role in pathologic fibroblast activation as shown above, it can be a promising therapeutic target for various fibroblast-related diseases. Indeed, we demonstrated the value of Twist1 as a new drug target in vivo using fibroblast-specific Twist1 knockout mice (FSP1-Cre; Twist1[fl/fl] mice) in this study (Figs. 6 and 7); In this experiment, FSP1-promoter Cre mice were employed to generate fibroblast-specific Twist1 deletion. Fibroblast-specific protein 1 (FSP1, also called S100a4) is expressed mainly in fibroblasts of various organs undergoing tissue remodeling including skin, lung and kidney. FSP-1 has recently been reported as a marker of inflammatory macrophages in liver injury but has been well studied as an improved marker of fibroblasts that could be useful in investigating the pathogenesis of cancer and fibrosis[46–49]. TNC has also recently been considered a promising drug target[50–52]. Furthermore, all three genes are absent in normal adult tissue, minimizing the possibility of fatal side effects. For instance, tamoxifen-mediated Twist1 deletion in adult tissue using the Cre-ER/loxP system did not reveal any fatal effects[53]. Because fibroblast activation is so crucial in both fibrotic diseases and cancer microenvironments, our study is expected to provide a new realm of therapeutic targets for both fibrosis and cancer.

In conclusion, we confirmed that Twist1, Prrx1, and TNC create a PFL in activated fibroblasts, and we inferred that this Twist1-Prrx1-TNC PFL operated as a bistable (ON/OFF) switch through mathematical modeling; this concept has been thoroughly demonstrated by in vitro, in vivo, and clinical studies.

Moreover, the sustained switching ON of the PFL in fibroblasts induced by exogenous TNC reproduced fibroblastic nodules in vivo similar to the nodules in the IPF. Our clinical study strongly suggests that the sustained activation of the Twist1-Prrx1-TNC PFL is associated with a variety of fibrotic diseases and cancer stroma. Initially, we postulated that fibroblast activation might be determined by a certain complex logic circuit operating as a decision maker. The Twist1-Prrx1-TNC PFL functioning as a bistable switch is highly likely to be an integral part of this logic circuit. Finally, our study provides a pathway to discovering such an entire logic circuit by offering concrete grounds and also suggests highly effective drug targets directly related to the core mechanism of fibrotic diseases and cancer microenvironments.

## Methods

**Immunohistochemistry and tissue samples**. A total of 306 tissue samples were obtained from patients with esophageal squamous cell carcinoma (ESCC) after ethical approval by the institutional review board of Samsung Medical Center (Seoul, Korea). All patients provided informed consent for tissue donation. Immunostaining for the Twist1 protein was performed using an anti-Twist1 monoclonal antibody (Abcam, Cambridge, UK). Tissue microarrays composed of 306 ESCC samples with clinical data including age, sex, tumor size, depth of invasion (T), nodal status (N), metastasis (M), overall survival (OS) and disease-free survival (DFS) were used to identify the clinical significance of the target genes' expression in patients with ESCC. Staging based on TNM classification was applied according to guidelines from the 2010 American Joint Committee on Cancer staging manual. Immunohistochemical staining was estimated according to our previous studies[54]. Sections on microslides were deparaffinized with xylene, hydrated using a diluted alcohol series, and immersed in 0.3% $H_2O_2$ in methanol to quench endogenous peroxidase activity. Sections were then treated with TE buffer (Tris 10 mM and EDTA 1 mM, pH 9.0) for antigen retrieval. To reduce nonspecific staining, each section was treated with 4% skim milk in PBS with 0.1% Tween 20 (PBST) for 30 min. Sections were then incubated with primary antibodies: anti-Twist1 (ab50887, 1:200), anti-Prrx1 (HPA051084, 1:200), and anti-Tenascin-C (ab108930, 1:500) in TBST containing 4% skim milk for 60 min at room temperature. After three successive rinses with a washing buffer, sections were then incubated with an anti-mouse/rabbit polymer kit (Envision Plus, Dako, Carpinteria, CA, USA) for 30 min at room temperature. The chromogen used was 3-amino-9-ethylcarbazole (AEC, SK-4205, Vector, Burlingame, CA, USA). Sections were counterstained with Meyer's hematoxylin and the virtual slide images were generated using Aperio® AT2 virtual slide scanner (Leica, Wetzler, Germany). As described in detail previously, immunohistochemical scores were measured semi-quantitatively[54]. In brief, we evaluated the staining intensity and the proportion of positive cells and then generated staining scores as follows: (Score 1), weak staining in <50% or moderate staining in <20% of stromal cells; (Score 2), weak staining in ≥50%, moderate staining in 20–50% or strong staining in <20%; (Score 3), moderate staining in ≥50% or strong staining in ≥20%.

**Statistical analysis**. Correlations were examined using Pearson's $\chi^2$, Fisher's exact test, or Spearman test, as appropriate. Wilcoxon rank sum test or $t$ test was used to evaluate differences between groups with continuous values. OS and DFS were determined using the Kaplan–Meier method and were compared using the log-rank test. Survival was measured from the date of surgery. The Cox proportional hazards model was used for multivariate analysis. Clinicopathologic factors, which were statistically significant in univariate analysis, were included as covariables in multivariate analysis. Hazard ratios (HR) and 95% confidence intervals (CI) were assessed for each factor. All tests were two sided, and $p$ value of less than 0.05 was considered statistically significant. The statistical analysis was performed using SPSS statistical software (SPSS Inc, Chicago, IL, USA).

**Gene expression data and gene set enrichment assay**. For TCGA data, the following type of "Level 3" processed and normalized gene expression data of 185 esophageal cancer patients (illuminaHiseq RNAseq V2 gene expression, RSEM normalized) were downloaded from the TCGA website. We selected 40 Twist1-high expressing group and 39 Twist1-low expressing group from 185 esophageal cancer patients included in TCGA dataset. Then to identify gene-signature-based differences between Twist1-high/low ESCC population, we performed GSEA using the Broad Institute's GSEA tool (http://www.broadinstitute.org/gsea/index.jsp).

**RNA extraction and quantitative real-time RT-PCR**. Total RNA was extracted using by RNeasy kit (#74104, Qiagen). cDNA was synthesized from 1 µg of total RNA using High capacity cDNA reverse transcription kits (#4368813, Applied Biosystems, Foster City, CA, USA), according to the manufacturer's protocol. Real-time RT-PCR was performed using the ABI 7900 HT Fast Real-Time PCR system (Applied Biosystems, Foster City, CA). GAPDH was used as an internal loading

control. PCR was conducted using Sybrgreen PCR master mix (Applied Biosystems). Thermal cycling conditions comprised 1 step for 10 min at 95 °C followed by 40 cycles at 95 °C for 15 s and at 60 °C for 1 min. Each measurement was performed in triplicate. PCR product quality was monitored using post-PCR melt-curve analysis. Cycle threshold (Ct), the fractional cycle number at which the amount of amplified target reached a fixed threshold, was determined. The primer sets are described in Supplementary Table 2.

**Western blotting analysis**. Cell lysates were isolated by RIPA lysis buffer (50 mM Tris pH 7.4, 150 mM NaCl, 1 mM EDTA, 1% Triton x-100, 1% Na-Doc, 0.1% SDS) supplemented with protease inhibitor cocktail (Roche). Cell extracts were quantitated using a BCA protein assay kit (Thermo). Western blot analysis was performed standard techniques for Twist1 (Abcam, ab50887, 1:1000, Cambridge, MA), Prrx1 (Origene, TA803116, 1:1000, Rockville, MD, USA), TNC (Genetex, GTX62552, 1:1000, USA) and alpha-tubulin (Santa Cruz, TU-02, 1:3000, Dallas, TX). Protein detection for Western blotting was performed using ECL reagent (#1705061, Bio-Rad). The antibodies are described in Supplementary Table 1. Uncropped images of the gels are shown in Supplementary Fig. 11.

**Cell lines**. Normal human lung fibroblasts IMR90 and MRC5 were obtained from the American Type Culture Collection (ATCC) and cultured in MEM (Invitrogen) supplemented with 10% fetal bovine serum (Invitrogen, Carlsbad, CA, USA) and 1× antibiotics (Invitrogen 15240-062, Gibco, containing 100 units penicillin and 100 µg of streptomycin per ml). The cultured cells were maintained at 37 °C under 5% $CO_2$ in a humid atmosphere. Cell identity was confirmed by STR (short tandem repeats) genotyping. Cells were tested for mycoplasma contamination once every month.

**Ex vivo culture**. Human esophagus and stomach tumor specimens were obtained from patients who were undergoing surgery at Samsung Medical Center of SungKyunKwan University of Medicine, Seoul, Korea. All patients provided informed consent for tissue donation and experimental uses were approved by the institutional review board of Samsung Medical Center. An experienced pathologist grossly examined and obtained representative samples of the tumor tissues (human stomach cancer-associated fibroblasts (SCAFs) and human esophageal cancer-associated fibroblasts (ECAFs)) and distal normal tissues (human stomach normal fibroblasts (SNFs)). The numbers after # are the order of the primary culture. Detailed maintenance procedures are described in our previous study[13].

Mouse dermal fibroblasts were isolated from fresh dorsal area tissue in 7-week-old wild-type mice; the tissues were cut into small pieces and minced with scalpels in culture dishes. We enzymatically dissociated the tissues in 20 ml of D/F12+ serum media containing collagenase I in a 37 °C incubator for 12 h using an orbital shaker. After digestion, the samples were centrifuged at 700 rpm for 5 min to separate the epithelial cells and fibroblasts. Fibroblasts were centrifuged at 800 rpm for 8 min; they were washed twice with PBS and cultured in D/F12 media supplemented with 10% FBS and 1% antibiotics.

**Lentivirus transduction**. Human embryonic kidney 293T cells were used for production of lentivirus. CAF and NF cells were transduced with lentivirus-expressing human pHR.CMV.Twist1.IRES.Hygro, pHR.CMV.Prrx1.IRES.Hygro and pHR.CMV.FLAG.IRES.Hygro control vector. The cells were infected with lentivirus and selected with hygromycin (Sigma-Aldrich). Human Twist1 Promoter-specific (~2260~+8)-GFP expression vector construct was generated by insertion of human Twist1 promoter sequence. The packaging of vector was obtained by transfection of 293T. CAF cells were transduced with pLVeGFP. hTwist1 (hTwist1 promoter GFP lentivirus) and pLVeGFP.Basic control. The cells were infected with lentivirus and selected by 0.5 µg/ml puromycin. Specific shRNAs were designed by using bioinformatics tools publicly available from MISSION shRNA, Sigma-Aldrich website. We designed sequences for efficient Twist1, Prrx1, and TNC targeting by shRNA. The ShRNA sequences are described in supplementary table 2. After virus transduction, cells were selected with 1 µg/ml puromycin. pCMV6-human TNC expression vector was purchased from Origene Technologies (cat no. RC215251).

**Tet-on system**. Cells were transduced with lentivirus-expressing Tet-on-advanced (Tet-on-inducible system, Clontech) and then cells were selected by 200 µg/ml of G418. These Tet-on advanced cells were infected again lentivirus containing with PLVX.Tight.Twist1.puro vector or FLAG control vector. These cells were cultured in a medium that either did or did not contain doxycycline (Clontech) or not supplemented with 10% of Tet-system-approved FBS (Clontech), a tetracycline-free serum developed to be optimal for the tetracycline-controllable expression system. After treatment of 100 ng of doxycycline, Tet-inducible Twist1-over-expression was confirmed.

**Chromatin immunoprecipitation assay**. CAF cells were grown to 70–80% confluency and fixed with 1% formaldehyde. Nuclear extracts were sonicated to shear the DNA to a length under about 500 bp. Twist1-DNA complexes were immuno-precipitated using Protein G Dynabeads (Invitrogen) conjugated with anti-Twist1

IgG (Abcam, ab50887) or control normal mouse IgG (sc2025, Santa Cruz). After the precipitated DNA was eluted, crosslinks were reversed by 0.3 M NaCl and purified by QIAquick PCR Purification Kit (Qiagen, CA, USA). Then PCR was performed on the purified DNA using primer sets are described in Supplementary Table 2.

**Luciferase assay and site-directed mutagenesis**. pGL3.Tenascin-C and PGL3. Prrx1 promoter luciferase construct sets were generated by insertion of TNC promoter sequence amplified by each primer sets. HT1080 (Fibrosarcoma) cells ($5 \times 10^4$ cells per well) were plated into 24-well plates and incubated for 24 h. Promoter vectors were cotransfected with Twist1 and Prrx1 expression vector (pSG5.HA vector was used as a control) into cells using Lipofectamine 2000 (Invitrogen). Luciferase reporter gene assays were performed using the Luciferase assay system (Promega). Twist1 binding of the E-BOX region on the Prrx1 promoter was mutated using pGL3.Prrx1 luciferase vector as the template DNA. PCR amplification for site-directed mutagenesis was performed using Prrx1 E-box mutant primers. Primer sets are described in Supplementary Table 2.

**FACS sorting**. CAF cells were infected with lentivirus containing human-Twist1 promoter-specific GFP reporter. FACS sorting was performed to isolate FITC population. GFP high (Twist1+) and GFP Low (Twist1−) cells were sorted by using FACS Aria III.

**Apoptosis assay**. Cells were induced apoptosis by treatment of 1.0 μM Doxorubicin. Apoptosis assay was performed according to the manual of BD Annexin V-APC apoptosis detection Kit (Cat. 550474, BD Biosciences, Franklin Lakes, NJ, USA). Briefly, cells were washed twice with PBS and then resuspended in 1× Annexin binding Buffer. These cells were stained with Annexin V conjugated with APC at room temperature in the dark for 20 min. Stained cells were detected by flow cytometry.

**Cutaneous wound healing model**. Adult wild-type (WT) mice (BALB/c) are used for these studies. The wound-healing studies were performed in accordance with a protocol approved by the institutional Animal Care and Use Committee of Laboratory Animal Research Center at Samsung Biomedical Research Institute. Four 5-mm biopsy punch skin wound were placed on the side of dorsal of mice. The wound was covered with a transparent semi-occlusive dressing (Tegaderm, 3 M, Saint Paul, MN, USA) to prevent desiccation. Wounds were observed until closure or animals were killed. The excisional wounds are excised on days 1, 3, 5, 9, 11, and 24. The wound areas were measured by using histological analysis. Investigators conducting functional measurements were blinded to the treatment group during data collection. We confirmed that the sample size is statistically different based on previously published experiments.

**Mathematical modeling**. We developed a mathematical model of the PFL composed of Twist1-Prrx1-TNC. When we construct this molecular regulatory network model, we did not attempt to describe the exact in silico replica of all biochemical species and their interactions. Rather, we constructed a minimal essential model of the PFL by including only the three major components and their regulations (see Supplementary text for further details on the mathematical model). Our model consists of three state variables and 16 kinetic parameters. The kinetic parameter values were estimated based on our own time course measurements by using the genetic algorithm implemented in Matlab ToolboxTM. Parameter estimation was carried out based on maximum likelihood estimation, starting from random initial guess to find the optimal parameter estimates that minimize the difference between experimental data and simulation data produced by the mathematical model (see Supplementary text for details).

**Mice model**. Mice with conditional deletion of Twist1 were generated by crossing FSP1-Cre-expressing mice (FSP1-Cre from Jackson Laboratory, Bar Harbor, ME, stock no. 012641)[55] with Twist1$^{fl/fl}$ (Twist1 flox/flox mice from MMRRC, stock no. 16842). Twist1$^{fl/fl}$ mice were backcrossed to a more tenth-generation congenic Balb/c background. FSP1-Cre;Twist1$^{fl/fl}$ mice, which have a fibroblast-specific Twist1 Knock-out (Twist1 K/O). 10 μg of Recombinant TNC (Millipore Cat No. CC065) or PBS was injected in wound area at 3−5 days after 5-mm biopsy punch skin wound. The mice were sacrificed after remodeling phase onset of excisional skin wound.

**Sequential IHC and image acquisition**. Sequential IHC was performed with 4 μm of FFPE tissue sections. Following deparaffinization, sections were rehydrated and stained with hematoxylin (S3301, Dako) for 1 min, cover-slipped with a Clear-Mount™ Mounting Solution (008010, Invitrogen). Tissue scan was proceeded using an Aperio ImageScope (Leica Biosystems). Slides were de-coverslipped in water for 1 h and subjected to endogenous peroxidase blocking followed by heat-mediated antigen retrieval with Tris-EDTA buffer (pH 9.0) for 15 min. Sequential IHC (staining, scanning and antibody stripping) was performed according to a protocol on previous report[56,57]. Slides were blocked with 4% skim milk in TBST for 30 min and incubated in primary antibodies: anti-Twist1 (ab50887, 1:200), anti-Prrx1 (HPA051084, 1:200), and anti-TNC (ab108930, 1:500) for 60 min at room temperature. Secondary antibody that was required to be compatible with the primary

host was incubated for 1 h, and protein was subsequently detected using the DAKO-Envision Dual Link Labelled Polymer (Anti-Rabbit) (#K5007, Dako Botany, NSW, Australia) for 30 min and the ImmPACT NovaRed Peroxidase Substrate Kit (#SK-4805, Vector Laboratories, Burlingame, CA, USA) for 3–4 min at room temperature.

**Image visualization and quantification**. Image analysis is consisted of an image processing, visualization and image quantification.

Each processing step and detailed protocol were described in a previous report[56]. First, images were coregistered for single cell-level overlapping using a CellProfiler v2.1.1, Alignment Batch.cppipe. pipeline.(free software, available at http://github.com). Next, image visualization was performed by converting the coregistered images into pseudo-color images using an ImageJ Fiji software. In ImageJ Fiji, coregistered and exported images were processed with Plugin, color deconvolution for separating of DAB/AEC and hematoxylin staining signal (Twist1-green, Prrx1-red, TNC-yellow and hematoxylin-blue). Single cell-based quantification of intensity was performed using Cellprofiler v2.1.1 and CellID_FlowCyt 6.9.15.cpproj automated segmentation pipeline (free software, available at http://github.com). This used AEC-staining images for measurement of intensities. Hematoxylin staining image was used for cell segmentation. All pixel intensity and measurements were saved to a file format for image flow cytometry data analysis using De Novo FCS Express 6 software.

**Data availability**. The authors declare that all the data supporting the results of this study are available within the article and its supplementary information files and from the corresponding authors upon reasonable request.

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

## Acknowledgements

This work was supported by the National Research Foundation of Korea (NRF) grants funded by the Korean government (MSIP) (2016R1A5A2945889), the Ministry of Science and ICT (2017R1A2A1A17069642, 2015M3A9A7067220, 2013M3A9A7046303, and 2015R1A2A1A15054021) and by the KAIST Future Systems Healthcare Project from the Ministry of Science and ICT. It was also supported by a grant of the Korea Health Technology R&D Project through the Korea Health Industry Development Institute (KHIDI), funded by the Ministry of Health & Welfare, Republic of Korea (grant number: HI14C2517).

## Author contributions

S.-Y.Y., K.-W.L. and D.S. performed the major experiments, data acquisition and analysis; S.A. conducted additional experiments and collected data and analysis. K.-H.C. and S.-H.K. designed experiments, supervised the study, and wrote the manuscript.

## Additional information

**Competing interests:** The authors declare no competing interests.

