## [Peer Review File · Nature Communications]

Editorial Note: Several images have been redacted as indicated to protect copyright.

Reviewers' comments:

Reviewer #1 (Remarks to the Author):

In this work the authors used combined experimental studies and mathematical modeling to reveal a positive feedback loop formed by Twist1, Prrx1, and TNC dictates fibroblast activation. I found that this work is very interesting and thoroughly analyzed. Persistent fibroblast activation has been associated with various disease conditions such as cancer and fibrosis. These authors convincingly showed that the three factors work synergistically to form a bistable switch.

I have the following concerns that need clarification.

- 1) Figure 1a: Is the number in the label "ESOCAF#8" an index of the patient sample? Similar for other labels. Need to indicate in the method part or caption. The strong linearity between the expressions of Twist1 and TNC is rather surprising given the stochasticity typically observed in single cell data. The results in Fig. 6a are what I would expect. The authors need to give more details of their experimental and image analysis. Do they see similar strong correlation in other samples? Shouldn't we expect bimodal distributions?
- 2) Figure 1a: The fonts of the scattered plots are rather small to see clearly. What are the axis units? I assume that they use some reduced units relative to certain reference value.
- 3) Figure 1C: I am confused with this result. Given the positive feedback loop identified by the author, shouldn't one expect to see increased TNC promoter activity with added Twist1, even they Twist1 do not bind to the promoter directly (as evidenced from the CIP data). Is it because the timing of measurement so that Prrx1 has not been activated? The authors need to make it clear.

Reviewer #2 (Remarks to the Author):

This is a very interesting paper that builds logically on the earlier data published by this group. It studies a positive feedback loop that occurs in fibroblasts during their activation. Previously published work demonstrated correlations between the expression of Twist1 and tenascin-C in these cells, but here the authors show that Twist1 does not act directly on tenascin-C expression, but instead increases the expression of Prrx1, which in turns increases tenascin-C expression. The tenascin-C in turn elevates Twist1 expression via and integrin-mediated pathway, creating the feedback loop. These results could lead to the development of therapies to fight fibrosis resulting from chronic inflammation.

The results are novel, and a broad range of techniques ranging from immunoblotting to mathematical modeling are employed. The results are generally presented clearly.

My concerns are the following:

- 1) While the paper (ref 19) cited for tenascin-C expression during development and disease

is a classic (now nearly 30 years old), it would be helpful to reference a more contemporary review of tenascin-C biology. Ref 19 needs to remain given the importance of wound healing in this paper.

2) Fig 1A doesn't add much to the paper as intracellular labeling of ECM is problematic and may be unrelated to the functional extracellular protein. It can be deleted since Fig. 1B is so convincing. The same goes for Fig. 3G.

3) There are many high and low molecular weight alternatively spliced variants of tenascin-C, but the blots are trimmed and the standards aren't shown. Which variants are being studied? It does appear that there may be multiple bands on some, but not all of the blots (e.g., #14 and #32 in Fig. 1F may be different variants, and a faint doublet may be present in Fig. 5E). The different variants may have different functions and may signal through different receptors, so keep track of low vs high molecular weight variants will be helpful.

4) The antibody to beta3 integrin prevents Twist1 levels from increasing in the presence of tenascin-C. This is likely to be blocking signaling via alphaVbeta3, which is the only beta3 integrin shown to date to be a tenascin-C receptor. Do other ECM ligands of alphaVbeta3 upregulate Twist1? Are these ligands found in tumor stroma or in healing wounds? Could they also be part of the proposed feedback loop?

Reviewer #3 (Remarks to the Author):

Understanding the mechanisms that control fibroblast activation is key to get insight into the contribution of fibroblasts in pathological conditions. This study is therefore very timely. The authors propose the existence of a positive feedback loop (PFL) between Twist1, Prrx1 and Tenascin-C based on overexpression and downregulation studies in fibroblasts in culture, and they show the the expression of the three genes correlates with properties of activated fibroblasts. Through standard promoter analyses and chromatin immunoprecipitation they show that The activation of tenascin C by Twist is likely mediated by Prrx1, which in turn it is induced by Twist. Then they go ahead and generate a mathematical model to explain the PFL and the model seems to meet the data obtained in the cultured fibroblasts. The dynamics of activation of the three genes also seem to fit well with the model, that proposes that the PFL functions as a bistable switch. Finally, they analyze fibroblasts from different sources and look at databases and patient samples to find that the three are either found expressed at a very low (basal) or high level, compatible with the bistable switch. Thus, all the data seem to be compatible with the existence of the PFL involving these three genes and bistability. Nevertheless, it is worth noting that the relationships described in this study were already known. As such, Prrx was known to induce Tenascin C expression (McKean et al, J Cell Biol 2003); Prrx1 was known to be downstream of Twist (Bildsoe et al, Dev Biol, 2016) and the coexpression of Prrx and Twist had been already observed in cancer cells and patients (Ocana et al, Cancer Cell, 2012). The only novelty here is that the relationship between the three fits with the existence of a bistable switch that may be significant in fibroblast biology in pathological conditions.

However, this has not been proven or challenged in vivo.

Another weak point is that once the PFL is activated can lead to irreversibility in terms of the phenotype, mainly occurring in pathological conditions, but in fact, this is not what happens during wound healing, where activation resolves. Thus, as already discussed in the manuscript there has to be a negative regulator that represses the expression of these genes to control reversibility. This is not clear at all from the model and it is crucial, as it could help to understand how activated fibroblasts can be treated as a target in disease. Essentially, how the "irreversible" switch in phenotype can be reversible.

The main key question that remains open is how significant is this particular PFL in vivo.

In addition, Twist, Prrx1 and Tenascin could belong together or individually to other regulatory networks relevant to impose a particular phenotype and activity. Also, are Twist, Prrx1 and Tenascin equally potent to efficiently initiate the PFL? Can the three trigger the irreversible phenotypic switch? Are all the three equally suitable to be targeted to abrogate the fibroblast- activation in vivo?

Point-by point response to Reviewers' comments

Reviewers' comments:

Reviewer #1 (Remarks to the Author):

In this work the authors used combined experimental studies and mathematical modeling to reveal a positive feedback loop formed by Twist1, Prrx1, and TNC dictates fibroblast activation. I found that this work is very interesting and thoroughly analyzed. Persistent fibroblast activation has been associated with various disease conditions such as cancer and fibrosis. These authors convincingly showed that the three factors work synergistically to form a bistable switch.

I have the following concerns that need clarification.

1) Figure 1a: Is the number in the label "ESOCAF#8" an index of the patient sample? Similar for other labels. Need to indicate in the method part or caption.

● Response to Reviewer #1's comment – The number in the label

-> As reviewer#1 commented, we found that in some parts of this paper, ECAF#8 (esophageal cancer associated fibroblast #8) was incorrectly labeled ESOCAF#8, so we correctly identified the cell as ECAF#8 in all text and figures. We isolated ECAF#8 cell from human esophageal cancer patient tissue. #Number means the name of the cell we are studying, which is the number we created to maintain and manage the cell. Similarly, were isolated SCAF#14 and SCAF#32 cells from stomach cancer patient tissue by *ex vivo* primary culture, which we described in our previous study (Lee et al, *Cancer Res*, 2015); we mentioned this in Methods. Furthermore, for further study, mouse dermal fibroblasts were isolated and added this to Methods as well (shown below in "**Ex vivo culture**").

Ex vivo culture

Human esophagus and stomach tumor specimens were obtained from patients who were undergoing surgery at Samsung Medical Center of SungKyunKwan University of Medicine, Seoul, Korea. An experienced pathologist grossly examined and obtained representative samples of the tumor tissues (human stomach cancer associated fibroblasts [SCAFs] and human esophageal cancer associated fibroblasts [ECAF]) and distal normal tissues (human stomach normal fibroblasts [SNFs]). The numbers after # are the order of the primary culture. Detailed maintenance procedures are described in our previous study

Mouse dermal fibroblasts were isolated from fresh dorsal area tissue in 7-week-old wild-type mouse; the tissues were cut into small pieces and minced with scalpels in culture dishes. We enzymatically dissociated the tissues in 20ml of D/F12+serum media containing collagenase I in a 37°C incubator for 12hrs using an orbital shaker. After digestion, the samples were centrifuged at 700 rpm for 5 min to separate the epithelial cells and fibroblasts. Fibroblasts were centrifuged at 800 rpm for 8 min; they were washed twice with PBS and cultured in D/F12 media supplemented with 10% FBS and 1% antibiotics.

2) Figure 1a: The strong linearity between the expressions of Twist1 and TNC is rather surprising given the stochasticity typically observed in single cell data. The results in Fig. 6a are what I would expect. The authors need to give more details of their experimental and image analysis. Do they see similar strong correlation in other samples? Shouldn't we expect bimodal distributions?

● **Response to Reviewer #1's comment – The axis units and image analysis in Fig 1a**

-> As reviewer #1 commented, strong linearity between the expressions of Twist1 and TNC may seem not to be consistent with prediction by Twist1-Prrx1-TNC PFL (positive feedback loop). However, this seeming discrepancy can be resolved by several additional experiments and deep insights into the details. First, because of limitation in confocal IHC analysis, the only the small number ($n < 100$) of cells in restricted area were analyzed. Confocal IHC is a useful method for confirming expression per cell, however there is a limit to observing the overall distribution pattern in the whole population. In contrast, the conventional other analytic tools such as western blotting and flow-cytometry are capable of analyzing a large quantity of cells. In order to compensate this limit, the more number of cells were additionally analyzed again by confocal IHC. Indeed, a vague yet obvious bimodal distribution of Twist1 and TNC expression was emerged as shown in the figure below.

The Twist1 and TNC expressions in SCAF#39 were measured again using confocal microscopy (SCAF#39 : cancer-associated fibroblasts isolated from gastric cancer)

Additionally, FACS-based analysis of large cell counts (5×10^6) showed a clear bimodal distribution of Twist1, Prrx1 and TNC expressions in ex vivo cultured cancer associated fibroblasts (Supplementary figure 5).

Secondly, cell culture condition of confocal IHC is considerably different from conventional cell culture condition and exerts more or less effects on gene expression profile of cells. For example, the cells were cultured on glass slide for confocal IHC instead of ordinary plastic ware. And the cells are cultured for a short time (2~4 days) before analysis. According to our in vivo experiment (fig. 7), high-concentration of TNC above a certain threshold is required for establishing PFL. In this case, the incubation time is so short that the cells were more likely to fail to produce and accumulate in extracellular space enough TNC to exceed the threshold. The initial cell splitting and changing the culture media during this confocal IHC study removes the extracellular TNC, therefore some PFL-positive fibroblasts are expected to escape from PFL. And these former PFL-positive cells may not be able to re-establish PFL again because of the lack of time.

Third, it is necessary to measure the extracellular TNC as well as intracellular levels because TNC mainly functions as an extracellular matrix. However, the quantity of extracellular TNC cannot be determined using in vitro culture based confocal IHC. Nevertheless, intracellular TNC may reflect the extracellular TNC to some extent because there are no other types of cells such as epithelial cells to secrete TNC in vitro culture.

In regard to this issue (measurement of extracellular TNC), other reviewer (reviewer #2) suggested removing the confocal IHC images because there is already reliable data showing the correlation between Twist1 and TNC. Accordingly we removed the confocal IHC data in revised manuscript.

3) **Figure 1C:** I am confused with this result. Given the positive feedback loop identified by the author, shouldn't one expect to see increased TNC promoter activity with added Twist1, even they Twist1 do not bind to the promoter directly (as evidenced from the CIP data). Is it because the timing of measurement so that Prrx1 has not been activated? The authors need to make it clear.

● **Response to Reviewer #1's comment – Promoter assay and CIP data Fig 1c**

->As reviewer #1 commented, our result in Figure 1c can be seemingly confusing because we showed that Twist1 indirectly activates the TNC via Prrx1 in figure 3. However, as reviewer #1 have already mentioned such an indirect activation of TNC promoter by Twist1 did not occur in luciferase assay due to insufficient incubation-time.

According to our experiments, Twist1 requires an incubation time of at least 5 days to increase TNC promoter activity indirectly through Prrx1 activation (As shown figure 3 and supplementary figure 3). On the other hand, luciferase-reporter assay to determine the activity of a given promoter is usually performed within 24-48 hours post-transfection. Thus, such an indirect activation of TNC promoter by Twist1 cannot occur in luciferase assay due to insufficient incubation time.

In general, luciferase assay is a useful method for studying the direct transcriptional regulation by gene of interest through transient co-transfection with exogenous regulatory and target gene promoter luciferase vectors. While the primary objective of promoter-luciferase assay is to quantify the direct effect on promoter strength by genes of interest, the long-time incubation distorted the results by inducing the various confounding factors such as cellular overpopulation and accumulated lipofectamine toxicity. Furthermore, extended incubation was reported to even reduce luciferase signal (Cheung, Sylvia T, et al). In fact, most of luciferase assay protocols recommend that cells are assayed 24~48 hours post transfection except for genes that require over 48hrs post transfection to express the genes. In this present study, we harvested cells 24 hours post transfection and measured TNC promoter luciferase activity.

Reviewer #2 (Remarks to the Author):

This is a very interesting paper that builds logically on the earlier data published by this group. It studies a positive feedback loop that occurs in fibroblasts during their activation. Previously published work demonstrated correlations between the expression of Twist1 and tenascin-C in these cells, but here the authors show that Twist1 does not act directly on tenascin-C expression, but instead increases the expression of Prrx1, which in turn increases tenascin-C expression. The tenascin-C in turn elevates Twist1 expression via an integrin-mediated pathway, creating the feedback loop. These results could lead to the development of therapies to fight fibrosis resulting from chronic inflammation.

The results are novel, and a broad range of techniques ranging from immunoblotting to mathematical modeling are employed. The results are generally presented clearly.

My concerns are the following:

1) While the paper (ref 19) cited for tenascin-C expression during development and disease is a classic (now nearly 30 years old), it would be helpful to reference a more contemporary review of tenascin-C biology. Ref 19 needs to remain given the importance of **wound healing** in this paper.

● Response to Reviewer #2's comment – Reference paper in introduction

→ As reviewer #2 suggested, we added a reference paper to the revised manuscript: K. S. Midwood et al., "Advances in tenascin-C biology, *Cell. Mol. Life Sci.* (2011) 68:3175–3199 DOI 10.1007/s00018-011-0783-6." This is an in-depth review paper on TNC that covers its various roles, and it can be used to refer mainly to its function under pathologic conditions.

2) Fig 1A doesn't add much to the paper as intracellular labeling of ECM is problematic and may be unrelated to the functional extracellular protein. It can be deleted since **Fig.1B** is so convincing. The same goes for **Fig. 3G**.

● Response to Reviewer #2's comment – Intracellular labeling of ECM in Fig 1B

→ As the reviewer #2 recommended, Fig.1B and Fig.3G were deleted in the figures.

3) There are many high and low molecular weight alternatively spliced variants of tenascin-C, but the blots are trimmed and the standards aren't shown. Which variants are being studied? It does appear that there may be multiple bands on some, but not all of the blots (e.g., #14 and #32 in Fig. 1F may be different variants, and a faint doublet may be present in Fig. 5E). The different variants may have different functions and may signal through different receptors, so keep track of low vs high molecular weight variants will be helpful.

● Response to Reviewer #2's comment – Spliced variants of Tenascin-C

→ As the reviewer commented, there are various TNC isoforms with different molecular weights and different functions. This is one of the most interesting issues in fibroblast activation.

To begin with, we examined which kinds of TNC isoforms were expressed in fibroblasts using western blotting analysis. In cancer-associated fibroblasts derived from stomach cancer (SCAF#14 and SCAF#32), large-sized isoforms (210, 250, 350kDa) were predominant, whereas TNC-S (fully spliced-small) was not observed at all as shown below. In our previous report (Plos ONE, 2016), we identified that large-sized TNC isoforms (both 350Kda and 250Kda) were again predominant in various cancer associated fibroblasts isolated from stomach and esophageal cancer. Collectively, in all these CAFs (total 5, SCAF#14,#32,#36,#39 and ECAF#8), large sized TNC isoforms were predominant. Our results are also highly consistent with Yoshida et al.'s report in which large-sized TNC variants were preferentially expressed in stromal fibroblasts within cancer tissues. And Bhattacharyya et al. reported the expression of TNC large isoform in organ fibrosis. All these data consistently indicate that large-sized TNC isoforms are predominantly expressed in these activated fibroblasts.

Figure 1e. TNC expression in CAFs isolated from gastric cancer (TNC KD : TNC knock-down)

TNC expression in CAFs isolated from gastric and esophageal cancer (plos one 2016)

Another interesting finding is that Twist1 induced specifically large-sized TNC isoform only. Doxycyclin-dependent Twist1 induction in MRC5 (lung fibroblast) resulted in production of only large-sized TNC isoform (about 350kDa) without any effect on other isoforms of TNC as shown below.

We established doxycyclin dependent Twist1-inducible system in MRC5 (lung fibroblast). In response to doxycyclin treatment, only large isoform of TNC was increased after Day 5. This figure is presented as Figure 3e in this manuscript.

Based on these findings, we focused on the large isoform of TNC. Next, we studied the in vivo functional effect of large-sized TNC isoform on fibroblast activation. We injected a 250kda recombinant TNC (Millipore cat no. CC065) into the wounded areas of mice. This large-sized TNC isoform induced the persistent activation of fibroblast through Twist1-Prrx1-TNC activation, which resulted in formation of fibroblastic nodule similar to nodules in idiopathic pulmonary fibrosis. Most interestingly, small-sized TNC isoform failed to induce such fibroblast activation. We repeated same

experiments by injecting small-sized TNC isoform (90Kda, R&D cat no. 3358-TC) instead of large-sized TNC isoform and there was not any difference compared with PBS-treated controls. We newly added these *in vivo* results in Figs. 6 and 7 in revised manuscript.

In conclusion, our data strongly indicate that large-sized TNC isoforms have an ability to strongly activate the fibroblast. And Twist1-Prrx1-TNC PFL is highly likely to be mediated by large-sized TNC isoforms. The role of small-sized TNC isoforms (TNC-S) in fibroblast activation was not studied in present study; however TNC-S is speculated to have its own functions. Our data highlight the significance of TNC isoforms in regulation of fibroblast activation.

4) The antibody to beta3 integrin prevents Twist1 levels from increasing in the presence of tenascin-C. This is likely to be blocking signaling via alphaVbeta3, which is the only beta3 integrin shown to date to be a tenascin-C receptor. Do other ECM ligands of alphaVbeta3 upregulate Twist1? Are these ligands found in tumor stroma or in healing wounds? Could they also be part of the proposed feedback loop?

● **Response to Reviewer #2's comment – Other ECM ligands of alphaVbeta3**

→ As reviewer #2 recommended, we performed additional experiments to identify other ECM ligands of integrin aVb3 to upregulate Twist1. For this experiment, we first isolated the dermal fibroblasts of wild-type mice and seeded them in 6-well plates. Then we added various recombinant ECM protein ligands of aVb3 such as fibronectin, osteopontin, vitronectin, mmp2, and TNC into dermal fibroblasts, and we determined the Twist1 expression by western blotting analysis. The results were quantitatively analyzed using imageJ Fiji. As shown below, this western blotting analysis showed that TNC significantly increased Twist1 expression than any other ECM ligands

• **Treatment of various recombinant protein ligands of aVb3(Fibronectin, osteopontin, vitronectin, MMP2, and TNC) into mouse dermal fibroblasts.**

Next, we examined the correlation between these ECM ligands and Twist1 in pathologic fibroblasts. We assessed the mRNA expressional level of these ligand proteins in SCAF #32 after knockdown of twist1 expression. TNC was the only gene whose expression decreased when Twist1 expression decreased. Conversely, we examined the expression of ligand genes when the Twist1 was overexpressed in SNF #32 (isolated from the same patient with CAF #32) and found that Twist1 increased TNC and MMP2.

Treatment of various recombinant protein ligands of aVb3(Fibronectin, osteopontin, vitronectin, MMP2, and TNC) into mouse dermal fibroblasts.

The mRNA expressional levels of α v β 3 ligand proteins in SNF#32 after overexpression of Twist1. SNF#32 (normal fibroblast isolated from normal stomach tissue)

The mRNA expressional levels of α v β 3 ligand proteins in SCAF#32 after knock-down of Twist1. SCAF#32 (cancer-associated fibroblast isolated from stomach cancer tissue)

These results indicate that of these tested ECM ligands, TNC could be the only ECM protein that is regulated by Twist1. In our previous study (cancer research 2015:73-85), using the mRNA microarray and bioinformatical analysis, we investigated the genes whose expression was increased when Twist1 was upregulated and those with suppressed expression when Twist1 was downregulated in fibroblast. As a result, TNC is the only ECM protein that was enhanced by Twist1 as well as suppressed by silencing Twist1. All these data strongly suggest that TNC is very special ECM protein regarding Twist1-induced fibroblast activation.

[redacted]

In our previous study, we found that TNC was the only ECM protein that was significantly influenced by not only Twist1 induction but also knockdown of Twist1 in fibroblasts. (reference : Twist1 is a key regulator of cancer-associated fibroblast. Cancer Research 2015;75:73-85)

Reviewer #3 (Remarks to the Author):

Understanding the mechanisms that control fibroblast activation is key to get insight into the contribution of fibroblasts in pathological conditions. This study is therefore very timely. The authors propose the existence of a positive feedback loop (PFL) between Twist1, Prrx1 and Tenascin-C based on overexpression and downregulation studies in fibroblasts in culture, and they show the the expression of the three genes correlates with properties of activated fibroblasts. Through standard promoter analyses and chromatin immunoprecipitation they show that The activation of tenascin C by Twist is likely mediated by Prrx1, which in turn it is induced by Twist. Then they go ahead and generate a mathematical model to explain the PFL and the model seems to meet the data obtained in the cultured fibroblasts. The dynamics of activation of the three genes also seem to fit well with the model, that proposes that the PFL functions as a bistable switch. Finally, they analyze fibroblastse from different sources and look at databases and patient samples to find that the three are either found expressed at a very low (basal) or high level, compatible with the bistable switch. Thus, all the data seem to be compatible with the existence of the PFL involving these three genes and bistability. Nevertheless, it is worth noting that the relationships described in this study were already known. As such, Prrx was known to induce Tenascin C expression (McKean et al, J Cell Biol 2003); Prrx1 was known to be downstream of Twist (Bildsoe et al, Dev Biol, 2016) and the coexpression of Prrx and Twist had been already observed in cancer cells and patients (Ocana et al, Cancer Cell, 2012). The only novelty here is that the relationship bewteen the three fits with the existence of a bistable switch that may be significant in fibroblast biology in pathological conditions.

Another weak point is that once the PFL is activated can lead to irreversibility in terms of the phenotype, mainly occurring in pathological conditions, but in fact, this is not what happens during wound healing, where activation resolves. Thus, as already discussed in the manuscript there has to be a negative regulator that represses the expression of these genes to control reversibility. This is not clear at all from the model and it is crucial, as it could help to understand how activated fibroblasts can be treated as a target in disease. Essentially, how the “irreversible“ switch in phenotype can be reversible.

The main key question that remains open is how significant is this particular PFL *in vivo*.

In addition, Twist, Prrx1 and Tenascin could belong together or individually to other regulatory networks relevant to impose a particular phenotype and activity. Also, are Twist, Prrx1 and Tenascin equally potent to efficiently initiate the PFL? Can the three trigger the irreversible phenotypic switch? Are all the three equally suitable to be targeted to abrogate the fibroblast-activation *in vivo*?

[1] Response to Reviewer #3's comment – *in vivo* evidence for the PFL

→ As reviewer #3 commented, Twist1, Prrx1, and TNC could be members of other regulatory networks in fibroblasts; the complexity of regulatory networks is likely to be greater when fibroblasts are activated than when they are in a quiescent state. Thus, it is highly desirable to demonstrate the presence and significance of this PFL *in vivo*, especially in the activated state.

A. Predictions for *in vivo* model

In order to confirm the PFL *in vivo* and explore its significance in regulating fibroblasts, we examined its responses to stimuli that disturb certain elements of the loop; for instance, Nijenhuis et al showed the significance of their PFL *in vivo* by inducing or deleting key loop components. Using a similar method, we attempted to confirm our PFL *in vivo* by validating our PFL-derived predictions, which were as follows:

Prediction #1: Twist1, Prrx1, and TNC remain simultaneously positive in activated fibroblasts *in vivo*.

Prediction #2: Our mathematical model based on *in vitro* data predicts that if the feedback strength of TNC to Twist1 (TNC→Twist1) is sufficiently strong, the hysteresis can be extremely exaggerated and the fibroblast can be irreversibly activated.

B. Validating our hypotheses

We confirmed these predictions in the *in vivo* model by stimulating the activated fibroblasts and perturbing key parameters. We also challenged our PFL hypothesis by examining whether these predictions could be explained by the null hypothesis. In fact, findings for prediction #1 were already shown *in vivo* in Figure 6 in the revised manuscript and confirmed again in these additional *in vivo* experiments using multiplex immunohistochemistry. To verify prediction #2, we stimulated activated fibroblasts with high levels of TNC. Specifically, we activated fibroblasts *in vivo* by making excisional wound in the skin of the mice and then injecting TNC into the wounded tissue. Interestingly, high exogenous TNC induced sustained fibroblast activation, which resulted in the formation of the fibrotic nodules shown in Figure 6. However, this result can be also explained by null hypothesis #2, in which TNC can form a positive feedback loop with some gene X other than Twist1 or Prrx1 to show sustained activation, as shown below.

Various hypotheses about the relationship of the three genes.

Thus, to exclude null hypothesis #2, we injected TNC into the cutaneous wounds in fibroblast-specific Twist1 knockout mice rather than wild-type mice. For this, we generated fibroblast-specific Twist1 knockout mice by crossing the FSP1 (fibroblast-specific protein1)-cre mice with Twist1^{Flox/Flox} mice. Indeed, without Twist1, TNC alone fails to induce sustained fibroblast activation, as shown in Figure 6 in the revised manuscript. Therefore, we concluded that our hypothesis was confirmed not only because this finding was highly consistent with our hypothesis but also because it could not be

explained by any null hypothesis.

The important findings we gained from this *in vivo* experiment are as follows:

- ① High exogenous TNC created patchy fibrotic foci that persisted abnormally in wound healing tissue.
- ② TNC was expressed in overwhelmingly high levels in these patchy fibrotic foci, and Twist1 expression was confirmed only in fibroblasts surrounded by very high levels of TNC. Additionally, we only observed Twist1/Prrx1/TNC all-positive fibroblasts in these fibrotic foci.

C. Irreversible fibroblast activation *in vivo*

Despite TNC treatment, we did not observe any of these findings at all in the fibroblast-specific Twist1 knockout mice. The fibrotic foci created by exogenous TNC injection are very similar to those found in irreversible and perpetual fibrotic diseases such as idiopathic pulmonary fibrosis (IPF), as shown below. In IPF, these fibrotic foci are the strongholds of persistent fibrosis and so specific to IPF that pathologists use them as diagnostic criteria. One very interesting finding was that activated epithelial cells were accompanied in both fibrotic foci of IPF- and TNC-induced fibrotic nodules. This implies that crosstalk may play a critical role in abnormal fibroblast activation.

H&E staining of IPF patient's tissue and TNC- induced mouse fibrotic nodule .

These findings strongly suggest that high exogenous TNC above a certain threshold can induce irreversible fibroblast activation through a Twist1-Prrx1-TNC PFL. Next, we succeeded in reproducing this phenomenon in an *in vitro* system for further analysis, and indeed, high exogenous TNC above a certain threshold induced abrupt changes in Twist1 expression in *in vitro* fibroblast cultures. Based on the quantitative data derived from our *in vitro* study, we explored the key mechanism behind this phenomenon using mathematical modeling (Fig. 7b in the revised manuscript). This modeling indicated that the feedback component of TNC→Twist1 follows Michaelis-Menten kinetics: If $V_{\text{TNC} \rightarrow \text{Twist1}}$ of the Michaelis-Menten equation is above a certain level and K_m is below the threshold simultaneously, the high level of TNC can function as a switch to induce abrupt transition from reversible to irreversible fibroblast activation (see Fig. 7c for detailed explanation).

[redacted]

D. Clinical study

As shown above, through the *in vivo* model study, we confirmed that the Twist1-Prrx1-TNC PFL functions as a main axis in fibroblast activation during wound healing. Furthermore, we found that exogenous TNC above a certain level can induce irreversible and pathologic fibroblast activation through the Twist1-Prrx1-TNC PFL. These results are again strongly supported by our clinical studies, which can be considered an independent set of *in vivo* studies. These clinical studies are summarized as follows:

- (1) The co-expression of the Twist1/Prrx1/TNC PFL was intensively and highly selectively identified in a cancer-associated fibroblast with a strong correlation with poor prognosis. (Figure 8 and supplementary figure 8)
- (2) The co-expression of the Twist1/Prrx1/TNC PFL was characteristically detected in the fibrotic foci of various fibrotic diseases such as idiopathic pulmonary fibrosis, keloid disease, and desmoid tumor (aggressive fibromatosis; Figure 8 and Supplementary Figure 8)

E. Discussion

Our *in vivo* findings are corroborated by several other researchers' *in vivo* studies, including that by Bhattacharyya et al, who demonstrated that TNC was required for skin and lung fibrosis using TNC knock-out mice; however they did not show *in vivo* TNC's potential to induce fibrosis directly. These authors also suggested TNC's role as ligand as a toll-like receptor without *in vivo* evidence, and Fu et al reported that TNC provides a special microenvironment for fibrosis (fibrogenic niche) in kidney fibrosis. Recently, Palumbo-Zerr et al demonstrated that fibroblast-specific deletion of Twist1 attenuated skin fibrosis using conditional knock-out mice. (col1a2-creER :: Twist1^{loxp/loxp} mouse)

Although these authors' data strongly support our result, their data are all fragmented and fail to show a big picture of fibroblast activation and fibrosis. Furthermore, this big picture needs to contain positive feedback in any form because the dramatically rapid fibroblast activation observed in early phase of wound healing is impossible without positive feedback loops: We are the first to present the big picture with *in vivo* support for it. Additionally, TNC is known to act as a fibrogenic niche but its detailed mechanism is unknown. In this study, we revealed the key mechanism of TNC by confirming *in vivo* that its action was completely blocked when Twist1 was deleted.

[2] Reply to Reviewer #3's comment – Negative regulator for the PFL

→ As the reviewer pointed out, an additional negative regulator to terminate the Twist1-Prrx1-TNC PFL should be introduced to explain the reversibility of PFL during wound healing. Indeed, our wound healing experiment revealed that the Twist1, Prrx1, and TNC protein levels abruptly decreased after the proliferative phase as shown below, and we discussed this issue in the manuscript. Therefore, we now hypothesize that there may exist a negative regulator, X, that starts to operate when the PFL is highly activated and then suppresses the loop. To investigate the role of such a possible negative regulator, we further explored the literature, and we found that TNC is temporally expressed during embryogenesis and wound healing but undetectable in most normal tissues.

Western blot analysis of Twist1 and TNC in mouse wound tissue by date after wound healing assay

By contrast, persistent activation of TNC occurs under a variety of pathological conditions such as tissue fibrosis and cancer. These findings imply that TNC should normally be under tight regulation and can be an important mediator for the hyper-activation of the PFL in the absence of such regulation. TNC is degraded by the matrix metalloproteinase (MMP) family (Siri, 1995), and overexpression of both TNC and MMPs often occurs under pathological circumstances such as tumorigenesis (Mackie, 1997), which suggests that there should be a negative feedback loop between TNC and MMPs. Moreover, RhoA-mediated signaling activated by mechanical stress in fibroblasts induces TNC expression, but exogenous TNC interferes with RhoA activation and ultimately suppresses TNC transcription, which suggests the existence of a negative feedback loop (NFL) that controls TNC expression (Imanaka-Yoshida, 2014). Therefore, we modeled Twist1-Prrx1-TNC PFLs and NFLs mediated by TNC and X and determined the kinetic NFL parameters such that the hyper-activated PFL was attenuated (Supplementary Figure 9a). On stimulation with doxycycline, the levels of Twist1, Prrx1, and TNC increased until TNC reached a certain level (Supplementary Figure 9b). In the middle and late phases, however, the activity of these three molecules decreased owing to the activation of the negative regulator, X. These results are well in accord with the expression patterns of the three genes in fibroblasts (Figs. 5c and d), suggesting that the X-mediated NFL coupled with the Twist1-Prrx1-TNC PFL has an important role in reversibility during wound healing. Similar to the case when PFL is hyper-activated, impairment of this negative feedback loop can also lead to pathological conditions in which PFL (e.g., TNC) exhibits irreversible hysteretic behavior along with the increase followed by decrease of doxycycline (Supplementary Figures 9c and d). By further analyzing the hysteretic behavior of TNC, we can develop efficient strategies to restore the reversibility from the irreversible switches under pathological conditions. For instance, an irreversible state on the phase plane corresponding to the irreversibility of TNC can be converted to a reversible state by increasing the activity of X to reinforce the NFL and/or inhibiting the positive feedback strength (Supplementary Figure 9e). Therefore, the next challenge will be to uncover the underlying mechanism of this negative feedback regulation, which is crucial in both maintaining the homeostasis of fibroblasts during wound healing and controlling the irreversibility of the PFL switch under pathological conditions. Following the reviewer's comment, we have included this point in the Supplementary note of the revised manuscript.

Supplementary Figure 9. Hysteresis analysis of a coupled positive and negative feedback loop (A) Schematic diagram of the coupled Twist1-Prrx1-TNC PFL and NFL and the mathematical description of TNC and X. The state equations for Twist1 and Prrx1 are the same as those depicted in Fig. 4(a). (B) Temporal profiles of the fold changes of Twist1, Prrx1, TNC, and X under doxycycline stimulation (100ng). (C) and (D) Various profiles for the hysteretic TNC response to the doxycycline stimulation along with the increased negative (C) and positive (D) feedback strength. (E) Phase diagram illustrating the irreversibility of TNC with respect to the negative and positive feedback strength (see the detailed descriptions of the irreversibility of TNC in Fig. 4e).

[3] Reply to Reviewer #3's comment – Potential of each member of PFL to initiate PFL as well as therapeutic target

→ As reviewer #3 commented, it is necessary to validate each member of the PFL to initiate the loop. In this study, we already demonstrated the potential of TNC to initiate the Twist1-Prrx1-TNC PFL *in vivo* (Fig. 6), and furthermore, exogenous TNC above a certain threshold triggered the phenotypic switch *in vivo* (Fig. 7). To show the potential of Twist1 and Prrx1 *in vivo*, fibroblast-specific inducible transgenic mice are required. For this, we are now making “col1a2-promoter rtTA transgenic mice” and have already established both “TRE-Twist1” and “TRE-Prrx1” mice. Meanwhile, we confirmed that both Twist1 and Prrx1 are indispensable for initiating the PFL using fibroblast-specific conditional knock-out mice. Specifically, we generated fibroblast-specific Twist1/Prrx1 knock-out mice by combining CRE-Loxp and CRISPR-CAS9 technologies, and then we evaluated the effects of deleting these genes on fibroblast activation during wound healing *in vivo*; we found that fibroblast activation was significantly impaired in the absence of either Twist1 or Prrx1 (data not shown). In contrast, we revealed that Twist1 deletion suppressed the irreversible fibroblast activation induced by hyper-activation of the Twist1-Prrx1-TNC PFL *in vivo* (Figure 6). This result strongly suggested the potential of Twist1 as a therapeutic target to turn off the irreversible phenotype switch in fibroblasts. One interesting fact is that there are no fatal side effects when both Twist1 and TNC are deleted after birth using tamoxifen-inducible knock-out mice. These results again show that Twist1 and TNC are very attractive therapeutic targets.

[4] Summary of our reponse to Reviewer #3's comment

- First of all, we expressed sincere gratitude for critical comments of reviewer #3. Thanks to reviewer #3's comments, this paper is significantly improved.
- We performed a series of genetically engineered mice based *in vivo* experiments to provide *in vivo* evidence for Twist1-Prrx1-TNC PFL (positive feedback loop).
- Through the *in vivo* experiments we found several novel and exciting discoveries.
 - [1] We succeeded in inducing the persistent activation of fibroblasts *in vivo* by triggering an irreversible activation of Twist1-Prrx1-TNC PFL.
 - [2] In particular, the persistent activation of fibroblast activation led to the formation of fibroblastic nodule whose histological characteristics are very similar to that of idiopathic pulmonary fibrosis.
 - [3] We confirmed again that this perpetual activation of fibroblast did not occur when Twist1-Prrx1-TNC PFL was disrupted *in vivo* using fibroblast-specific Twist1 knock-out mice .
 - [4] This phenomenon was successfully explained by computational simulation based on our mathematical model.
- These *in vivo* discoveries were supported by our clinical studies that Twist1-Prrx1-TNC PFL is strongly and specifically expressed in cancer associated fibroblast and abnormal fibroblasts of fibrotic diseases.
- All these *in vivo* data were added as Figure 6 and Figure 7.

References

1. Mackie, E.J. Molecules in focus: tenascin-C. *The international journal of biochemistry & cell biology* **29**, 1133-1137 (1997).
2. Midwood, K.S., Hussenet, T., Langlois, B. & Orend, G. Advances in tenascin-C biology. *Cellular and molecular life sciences : CMLS* **68**, 3175-3199 (2011).
3. Nijenhuis, T. *et al.* Angiotensin II contributes to podocyte injury by increasing TRPC6 expression via an NFAT-mediated positive feedback signaling pathway. *Am J Pathol* **179**, 1719-1732 (2011).
4. Lee, K.W., Yeo, S.Y., Sung, C.O. & Kim, S.H. Twist1 is a key regulator of cancer-associated fibroblasts. *Cancer Res* **75**, 73-85 (2015).
5. Bhattacharyya, S. *et al.* Tenascin-C drives persistence of organ fibrosis. *Nat Commun* **7**, 11703 (2016).
6. Palumbo-Zerr, K. *et al.* Composition of TWIST1 dimers regulates fibroblast activation and tissue fibrosis. *Ann Rheum Dis* (2016).
7. Siri, A. *et al.* Different susceptibility of small and large human tenascin-C isoforms to degradation by matrix metalloproteinases. *The Journal of biological chemistry* **270**, 8650-8654 (1995).
8. Imanaka-Yoshida, K., Yoshida, T. & Miyagawa-Tomita, S. Tenascin-C in development and disease of blood vessels. *Anatomical record (Hoboken, N.J. : 2007)* **297**, 1747-1757 (2014).
9. Cheung, S.T., Shakibakho, S., So, E.Y. & Mui, A.L. Transfecting RAW264.7 Cells with a Luciferase Reporter Gene. *Journal of visualized experiments : JoVE*, e52807 (2015).
10. Yoshida, T., Akatsuka, T. & Imanaka-Yoshida, K. Tenascin-C and integrins in cancer. *Cell adhesion & migration* **9**, 96-104 (2015).

REVIEWERS' COMMENTS:

Reviewer #1 (Remarks to the Author):

As mentioned in my previous review, this is a comprehensive study that may have significant impact on cancer and fibrosis research. I notice that the authors had reexamined their confocal image and indeed there was artifact in their original data due to insufficient sampling. They have removed the results and the present results are sufficient to support their conclusion.

The manuscript can be improved if the authors seek some professional editing help, esp. on the language. They should also make sure that they provide explanation on the terms, esp. acronyms in their first usage. Below I just list a few examples.

Figure 1: The word "reciprocally" may cause confusion. I checked dictionary and talked with other colleagues. The word can both mean "mutual" and "inversely proportional". So I suggest to remove this word.

Figure 1a: I can't find explanation of acronym "SNF", and no explanation on what the authors refer to the columns "GFP", "Twist1" , "NS", etc. Some clear explanation in the caption will be helpful. For example, I assume ShTw1 refers to transfection with Twist1 shRNA. Don't assume a reader knows your previous paper.

Figure 5a: What are the inserts?

Figure 5b: What do the dashed circles and arrows refer to?

Figure 5d: Are these plots of data fro panel b?

Figure 9: They label as "Bistable swiching" and "Irreversible switching". Both are bistable switching, while the left is reversible and the right is irreversible.

Any idea on why the Hill coefficient n_1 is so big? This is a minor point without much concern from me. I understand that authors just fit the data. Parameters like n_1 may not be so sensitive for the fitting quality. That is, a smaller value may also give a good fit.

Reviewer #2 (Remarks to the Author):

The authors have done a fine job addressing the concerns that I had with the original submission.

Reviewer #3 (Remarks to the Author):

The authors have put much effort into this revised version. I appreciate very much the new data included, which are of high quality. This is a very interesting study

Only a minor point--just to remark that the Fsp-1 driver is not specific for fibroblasts. In fact, it also targets macrophages if not more cell types (Osterreicher et al, PNAS 2011). I suggest to include this caveat as a comment in the text. Provided the authors do that, given that there are no much better specific drivers available, I am very positive about this work.

Response to the Reviewer's Comments and Summary of Changes

Manuscript ID: NCOMMS-17-08836A

Title: The positive feedback loop formed by Twist1, Prrx1, and Tenascin-C bi-stably activates fibroblasts

Authors: S.Y. Yeo, K.-W. Lee, D. Shin, S. An, K.-H. Cho, and S.-H. Kim

Response to the specific comments of Reviewer 1:

As mentioned in my previous review, this is a comprehensive study that may have significant impact on cancer and fibrosis research. I notice that the authors had reexamined their confocal image and indeed there was artifact in their original data due to insufficient sampling. They have removed the results and the present results are sufficient to support their conclusion.

The manuscript can be improved if the authors seek some professional editing help, esp. on the language. They should also make sure that they provide explanation on the terms, esp. acronyms in their first usage. Below I just list a few examples.

[COMMENT #1] Figure 1: The word “reciprocally” may cause confusion. I checked dictionary and talked with other colleagues. The word can both mean “mutual” and “inversely proportional”. So I suggest to remove this word.

[RESPONSE] Thanks for the reviewer#1's detailed instructions. We are very sorry for causing confusion by using incorrect word. We agree with the reviewer#1's comment. As suggested by reviewer#1, we removed the word 'reciprocally' from Figure 1. We thank the reviewer#1 for pertinent comments.

[COMMENT #2] Figure 1a: I can't find explanation of acronym “SNF”, and no explanation on what the authors refer to the columns “GFP”, “Twist1” , “NS”, etc. Some clear explanation in the caption will be helpful. For example, I assume ShTw1 refers to

transfection with Twist1 shRNA. Don't assume a reader knows your previous paper.

[RESPONSE] We thank the reviewer#1 for helping us improve the clarity of our manuscript.

We carefully reviewed our manuscript and agree with the reviewer#1's comments that the explanation of acronyms is lacking in Figure 1. We have changed the Figure 1 legends, as follows:

(a) Induced Twist1 expression increased both the protein and mRNA of TNC in normal fibroblasts. Twist1 knock-down also decreased the TNC expression in CAFs.

(a) Normal fibroblasts were transduced with Twist1 encoding lentivirus or GFP control. Induced Twist1 expression increased both the protein and mRNA of TNC in stomach normal fibroblasts (SNF#14 and SNF#32) and esophageal normal fibroblast (ENF#8). CAFs were transduced with lentivirus expressing shRNA specific for Twist1 (shTw1) or non-specific shRNA (shNS). Knock-down of Twist1 also decreased the TNC expression in the esophageal and stomach cancer-derived CAFs (ECAAF#8, SCAF#14 and SCAF#32).

[COMMENT #3] Figure 5a: What are the inserts?

[RESPONSE] We are very sorry for the confusion caused by lack of explanation. In Figure 5a, The insert images are enlarged picture of stained cells in each staining figures. We have added a description of this to the legend section as follows.

(a) Twist1 expression was induced in fibroblasts (MRC5) using a doxycycline-mediated induction system, and we examined the impact of the induced Twist1 on the Prrx1 and TNC expression profiles using multiplex immunohistochemistry (mIHC). Bimodal distribution of the triple gene expression, such as all positive vs. all negative, was evident after 7 days of induction.

(a) Twist1 expression was induced in fibroblasts (MRC5) using a doxycycline-mediated induction system, and we examined the impact of the induced Twist1 on the Prrx1 and TNC expression profiles using multiplex immunohistochemistry (mIHC). Bimodal distribution of the triple gene expression, such as all positive vs. all negative, was evident after 7 days of induction. The insert images are enlarged picture of stained cells in each staining images.

[COMMENT #4] Figure 5b: What do the dashed circles and arrows refer to?

[RESPONSE] We are very sorry for the confusion caused by lack of explanation. In Figure 5b, The yellow box is a low magnification staining image of the mouse wound area. Dashed circle and arrow indicate Area of Interest (AOI) within yellow box. The AOI was shown in high power view to reveal the expression pattern of each genes in fibroblast.. We have added a description of this to the legend section.

(b) We examined the Twist1, Prrx1, and TNC expressions during cutaneous wound healing using multi-marker immunohistochemistry.

(b) We examined the Twist1, Prrx1, and TNC expressions during cutaneous wound healing using multiplex immunohistochemistry. The yellow box is a low power view of the mouse wound area. In this yellow box, dashed circle and arrow indicate Area Of Interest (AOI) which is shown as a high power view.

[COMMENT #5] Figure 5d: Are these plots of data fro panel b?

[RESPONSE] The plot is generated based on data derived from Figure 5b. We are very sorry for the confusion caused by lack of description. We have added a description of this to the legend section.

[COMMENT #6] Figure 9: They label as “Bistable switching” and “Irreversible switching”. Both are bistable switching, while the left is reversible and the right is irreversible.

[RESPONSE] As the reviewer pointed out, both are indeed bi-stable switching. In a normal condition (left), the levels of Twist1, Prrx1, and TNC completely return to their initial states for the removal of the doxycycline stimulus (see hysteretic responses of the three genes to the stimulus, Fig. 4c). However, in a pathological condition, the levels of three genes are sustained by highly activated positive feedback regulation and remain in their active states even after the stimulus is removed, which is easily understood by examining the increase of hysteresis width along with the increase of the feedback strength (Fig. 4d and e). Therefore, we have distinguished two cases by labeling the left as "Bi-stable switching" and the right as "Irreversible switching". To avoid any confusion, the terms were replaced by “Reversible bi-stable switching” and “Irreversible bi-stable switching” for normal and pathological conditions, respectively.

[COMMENT #7] Any idea on why the Hill coefficient n_1 is so big? This is a minor point without much concern from me. I understand that authors just fit the data. Parameters like n_1 may not be so sensitive for the fitting quality. That is, a smaller value may also give a good fit.

[RESPONSE] All the kinetic parameter values including Hill coefficients were estimated to fit our own time course data. To further examine the sensitivity of n_1 , we have tested whether our computational prediction (see Fig. 4f) was still valid for the smaller values of n_1 , $n_1 = 3$ and 5. As shown in the figure, we confirmed that the qualitative tendency of irreversibility of TNC was conserved for $n_1 = 3$ and 5.

[RESPONSE TO GENERAL COMMENT] In addition to these corrections as shown above, we have improved the manuscript through professional English editing service

Response to the specific comments of Reviewer 2:

[COMMENT #1] The authors have done a fine job addressing the concerns that I had with the original submission.

[RESPONSE] Thanks for reviewer#2's encouragement. We thank the reviewer#2 for the positive comments concerning our study.

Response to the specific comments of Reviewer 3:

The authors have put much effort into this revised version. I appreciate very much the new data included, which are of high quality. This is a very interesting study.

[COMMENT #1] Only a minor point--just to remark that the Fsp-1 driver is not specific for fibroblasts. In fact, it also targets macrophages if not more cell types (Osterreicher et al, PNAS 2011). I suggest to include this caveat as a comment in the text. Provided the authors do that, given that there are no much better specific drivers available, I am very positive about this work.

[RESPONSE] We thank the reviewer for this suggestion. FSP-1 is mainly considered a marker of activated fibroblasts. Recently, FSP-1 has been reported as a marker of inflammatory macrophages in liver injury. Therefore, we have now added comment and reference (Osterreicher et al, PNAS 2011) and also included several references of FSP-1 also identified fibroblasts in other organs including skin, lung and kidney (Strutz F, et al. 1995, Boye K, Maelandsmo GM 2010, Schneider M, et al. 2007). Furthermore, FSP-1 has been well studied as an improved marker for fibroblasts that could be useful for investigating the pathogenesis of cancer and fibrosis. As suggested by reviewer#3, we have added this mention in the revised Discussion section, as follows:

Fibroblast-specific protein 1 (FSP1, also called S100a4) is expressed mainly in fibroblasts of various organs undergoing tissue remodeling including skin, lung and kidney. FSP-1 has recently been reported as a marker of inflammatory macrophages in liver injury, but has been well studied as an improved marker of fibroblasts that could be useful in investigating the pathogenesis of cancer and fibrosis⁴⁶⁻⁴⁹.